



# Vertical and horizontal distribution of regional new particle formation events in Madrid

Cristina Carnerero[1,2], Noemí Pérez[1], Cristina Reche[1], Marina Ealo[1], Gloria Titos[1], Hong-Ku Lee[3], Hee-Ram Eun[3], Yong-Hee Park[3], Lubna Dada[4], Pauli Paasonen[4], Veli-Matti Kerminen[4], Enrique Mantilla[5], Miguel Escudero[6], Francisco J. Gómez-Moreno[7], Elisabeth Alonso-Blanco[7], Esther Coz[7], Alfonso Saiz-Lopez[8], Brice Temime-Roussel[9], Nicolas Marchand[9], David C. S. Beddows[10], Roy M. Harrison[10,11], Tuukka Petäjä[4], Markku Kulmala[4], Kang-Ho Ahn[3], Andrés Alastuey[1], Xavier Querol[1]

[1]Institute of Environmental Assessment and Water Research (IDAEA-CSIC), Barcelona, 08034, Spain.

[2]Department of Civil and Environmental Engineering, Universitat Politècnica de Catalunya, Barcelona, 08034, Spain.

[3]Department of Mechanical Engineering, Hanyang University, Seoul, Republic of Korea.

[4]Department of Physics, University of Helsinki, Helsinki, 00560, Finland.

[5]Centro de Estudios Ambientales del Mediterráneo, CEAM, Paterna, 46980, Spain.

[6]Centro Universitario de la Defensa de Zaragoza, Academia General Militar, Zaragoza, 50090, Spain.

[7]Department of Environment, Joint Research Unit Atmospheric Pollution CIEMAT, Madrid, 28040, Spain.

[8]Department of Atmospheric Chemistry and Climate, Institute of Physical Chemistry Rocasolano (IQFR-CSIC), Madrid, 28006, Spain.

[9]Aix Marseille Univ, CNRS, LCE, Marseille, 13003, France.

[10]National Centre for Atmospheric Science, University of Birmingham, B15 2TT, United Kingdom.

[11]Department of Environmental Sciences, Centre for Excellence in Environmental Studies, King Abdulaziz University, Jeddah, 21589, Saudi Arabia.

*Correspondence to*: Cristina Carnerero (cristina.carnerero@idaea.csic.es)

**Abstract.** The vertical profile of new particle formation (NPF) events was studied by comparing the aerosol size number distributions measured aloft and at surface level in a suburban environment in Madrid, Spain using airborne instruments. The horizontal distribution and regional impact of the NPF events was investigated with data from three urban and suburban stations in the Madrid metropolitan area. Intensive regional NPF episodes followed by particle growth were simultaneously recorded at three stations in and around Madrid, in a field campaign in July 2016. On some days a marked decline in particle size (shrinkage) was observed in the afternoon, associated with a change in air masses. Additionally, a few nocturnal nucleation mode bursts were observed in the urban stations, which could be related to aircraft emissions transported from the airport. Considering all simultaneous diurnal NPF events registered, growth rates were significantly lower at the urban stations, ranging 2.0-3.9 nm h$^{-1}$, compared to the suburban station (2.9-10.0 nm h$^{-1}$). Total concentration of 9.1-25 nm particles reached $2.8 \times 10^4$ cm$^{-3}$ at the urban station and $1.7 \times 10^4$ cm$^{-3}$ at the suburban station, the mean daily values being $3.7 \times 10^4$ cm$^{-3}$ ($2.2 \times 10^4$ cm$^{-3}$ at the suburban station) during event days. The formation rates of 9-25 nm particles peaked around noon and recorded a median value of 2.0 cm$^{-3}$ s$^{-1}$ and 1.1 cm$^{-3}$ s$^{-1}$ at the urban and suburban stations, respectively. The condensation and coagulation sinks presented minimum values shortly before sunrise, increasing after dawn reaching the maximum value at 14:00 UTC, with average daily mean values of $3.4 \times 10^{-3}$ s$^{-1}$ ($2.5 \times 10^{-3}$ s$^{-1}$ at the suburban station) and $2.4 \times 10^{-5}$ s$^{-1}$, respectively, during event days. The vertical soundings demonstrated that ultrafine particles (UFP) are transported from surface levels to higher levels, thus newly-formed particles ascend from surface to the top of the mixing layer. The morning soundings revealed the presence of a residual layer in the upper levels in which aged particles (nucleated and grown on previous days) prevail. The particles in this layer also grow in size, with growth rates significantly smaller than those inside the mixed layer. Under conditions with strong enough convection, the soundings revealed homogeneous




number size distributions and growth rates at all altitudes, which follow the same evolution in the other stations considered in this study. This indicates that NPF occurs quasi-homogenously in an area spanning at least 17 km horizontally. The NPF events extend over the full vertical extension of the mixed layer reaching as high as 3000 m. This can have consequences in the radiative balance of the atmosphere and affect the climate. Results also evidenced that total particle concentration in and

around Madrid in summer is dominated by NPF during summer, thus it may obscure the impact of vehicle exhaust emissions on levels of UFP.

## 1    Introduction

New particle formation (NPF) from gaseous precursors has been shown to cause high ultrafine particle (UFP) episodes in relatively clean atmospheres due to low condensation sinks (CS) originating from low pre-existing particle concentration

(e.g. Kulmala et al., 2000; Boy and Kulmala, 2002; Wiedensohler et al.,2002; Kulmala et al., 2004; Wehner et al., 2007; O'Dowd et al., 2010; Sellegri et al., 2010; Vakkari et al., 2011; Cusack et al., 2013a; Cusack et al., 2013b; Tröstl et al., 2016; Kontkanen et al., 2017). However, at mountain sites, precursors' availability seems to be the most influential parameter in NPF events, with higher values of CS during NPF events than during non NPF events (Boy et al., 2008; Boulon et al., 2010; García et al., 2014, Nie et al., 2014, among others). Tröstl et al. (2016) reported experimental results on nucleation driven by

oxidation of Volatile organic compounds (VOCs), and Kirkby et al. (2016) reported pure biogenic nucleation. Globally, and especially in urban areas, traffic emissions are a major source of UFP (Kumar et al., 2014; Ma and Birmili, 2015; Pey et al., 2008; Pey et al., 2009; Dall'Osto et al., 2012; Salma et al., 2014; Paasonen et al., 2016) and these arise from primary UFP exhaust emissions (Shi and Harrison, 1999; Shi et al., 2000; Charron and Harrison, 2003; Uhrner et al., 2012), condensation of semi-volatile phases vapor species that creates new UFP during dilution and cooling of engine exhaust emissions near the

source (Charron and Harrison, 2003; Kittelson et al., 2006; Robinson et al., 2007). Since these are formed very close to the source, most studies consider them as primary (e.g. Brines et al., 2015; Paasonen et al., 2016) or quasi-primary particles (Rönkkö et al., 2017). Other relevant UFP sources include industrial emissions (Keuken et al., 2015; El Haddad et al, 2013), city waste incineration (Buonanno and Morawska, 2015), shipping (Kecorius et al, 2016; Johnson et al., 2014), airports (Cheung et al., 2011; Hudda et al., 2014; Keuken et al., 2015) or even construction works (Kumar and Morawska, 2014).

NPF events contribute also significantly to ambient UFP concentrations in urban environments (Woo et al., 2001; Alam et al., 2003; Stanier et al., 2004; Wu et al., 2008; Costabile et al., 2009; Reche et al., 2011; Rimnácová et al., 2011; Salma et al., 2011; Salma et al., 2016; Harrison et al., 2011; Gómez-Moreno et al., 2011.; Wegner et al., 2012; von Bismarck-Osten et al., 2013; Dall'Osto et al., 2013; Betha et al., 2013; Cheung et al., 2011; Hussein et al., 2014; Liu et al., 2014; Salimi et al., 2014; Brines et al., 2014; Brines et al., 2015; Ma and Birmili, 2015; Minguillón et al., 2015; Hofman et al., 2016, Kontkanen

et al., 2017). Common features enhancing urban NPF are high insolation, low relative humidity, availability of $SO_2$ and organic condensable vapors, and low condensation and coagulation sinks (Kulmala et al., 2004; Kulmala and Kerminen, 2008, Sipilä, et al., 2010, Salma et al., 2016). Urban NPF episodes can be either regionally or locally driven and may or may not impact regional background areas (Dall'Osto et al., 2013; Brines et al., 2015; Salma et al., 2016). Cheung et al. (2011) and Brines et al. (2015) reported that in urban areas nucleation bursts without growth of particles are common; whereas the

frequently occurring 'banana like' nucleation bursts at regional background sites are scarcely detected at urban sites, probably because the high CS at traffic rush hours limits the duration of the particle growth. These processes seem to prevail in summer and spring in Southern European urban areas (Dall'Osto et al., 2013; Brines et al, 2014; Brines et al., 2015). Brines et al. (2015) also reported that in urban environments the highest $O_3$ levels occur simultaneously with NPF events, as well as the highest $SO_2$ concentrations and insolation, and the lowest relative humidity and NO and $NO_2$ levels. This close

association between $O_3$ and UFP may be due to ambient conditions that favor two different but simultaneous processes, or to the fact that they are two products of photochemical reactions in the same overall process.



Reche et al. (2011) evaluated the prevalence of primary versus newly formed UFP in several European cities and found a different daily pattern for the southern European cities, where the newly formed particles contributed substantially to the annual average concentrations, probably because of high insolation and possible site-specific chemical precursors. Brines et al. (2015) determined that NPF events lasting for 2 h or more occurred on 55 % of the days and those extending 4 h on 28 % of the days, being NPF the main contributor on 14-19 % of the time in Mediterranean and Sub-tropical climates (Barcelona, Madrid, Roma, Los Angeles and Brisbane). The latter percentages reached 2 % and 24-28 % in Helsinki and Budapest (Wegner et al., 2012 and Salma et al., 2016, respectively). Furthermore, Brines et al. (2015) calculated that 22 % of the annual average UFP number concentration recorded at an urban background site of Barcelona originated from NPF. Ma and Birmili (2015) reported that the annual contribution of traffic on UFP number concentration was 7, 14 and 30 % at roadside, urban background and rural sites respectively, in and around Leipzig, Germany. On the other hand, traffic emissions contributed 44-69 % to UFP concentrations in Barcelona (Pey et al., 2009; Dall'Osto et al., 2012, Brines et al., 2015), 65 % in London (Harrison et al., 2011; Beddows et al., 2015) and 69 % in Helsinki (Wegner et al., 2012).

Minguillón et al. (2015) and Querol et al. (2017a) demonstrated that intensive NPF episodes take place inside the planetary boundary layer (PBL) in Barcelona, occurring around midday at surface levels when insolation and dilution of pollution are at their maxima. Earlier in the morning NPF can only take place at upper atmospheric levels, at an altitude where pollutants are diluted, since at surface levels a high CS prevents particle formation.

While many studies have investigated NPF around the world, only a few have focused on the vertical distribution of these events (Stratmann et al., 2003; Wehner et al., 2010). In view of this, we devised a campaign with the aim to study photochemical episodes, including high $O_3$ levels and NPF in Madrid metropolitan area. In this work we will focus exclusively on the phenomenology of the NPF events, comparing the aerosol size distribution in surface level stations in urban and suburban environments, and aloft using airborne instrumentation with surface levels and studying its temporal evolution. In a twin article (Querol et al., 2017b) the study of the temporal and spatial variability of $O_3$ is presented.

## 2    Methodology

### 2.1    The study area

The Madrid Metropolitan Area (MMA) lies in the center of the Iberian Peninsula at an elevation of 667 m above sea level (a.s.l.). It is surrounded by mountain ranges and river basins that channel the winds in a NE-SW direction. Having an inland Mediterranean climate, winters are cool and summers are hot, and precipitation occurs mainly in autumn and spring. Road traffic and residential heating in winter are the main sources of air pollutants, with small contributions of industrial and aircraft emissions (Salvador et al., 2015).

In summer, the area is characterized by strong convection, which results in PBL heights as high as 3000 m above ground level (a.g.l.), and mesoscale recirculation caused by anabatic and katabatic winds in the surrounding mountain ranges (Plaza et al., 1997, Crespí et al., 1995), which can lead to accumulation of pollutants if the recirculation persists for several days.

Cold and warm advection of air masses associated with the passage of upper level troughs and ridges over the area give rise to a sequence of accumulation and venting periods, respectively. During accumulation periods, pollutants accumulate in the area and concentrations increase for 2-6 days, until a trough aloft brings a cold advection and a venting period starts. For a detailed description of the meteorological context during the campaign see Querol et al. (2017b).

A few studies have focused on NPF events in the area. For instance, Gómez Moreno et al. (2011) reported NPF episodes in Madrid to be 'not a frequent phenomenon', since only 63 events per year were detected, 17 % of the total days, occurring mostly in spring and summer. However, Brines et al. (2015) reported both intensive summer and winter NPF episodes at the





same station that accounted for 58 % of the time as an annual average, considering the prevalence of nucleation bursts during 2 hours or more. Alonso-Blanco et al. (2017) described the phenomenology of particle shrinking events, i.e. a decline in particle size caused by particle-to-gas conversion, at an urban station in Madrid (CIEMAT), stating that they occur mainly between May and August in the afternoon, due to either a change in wind direction or the reduction of photochemical

processes. Particle shrinkage following their growth is not a common phenomenon but has been observed in a few areas around the world. Yao et al. (2010), Cusack et al. (2013a and 2013b), Young et al. (2013), Skrabalova et al. (2015) and Alonso-Blanco et al. (2017) and references therein, reported shrinkage rates ranging from –1.0 to –11.1 nm h$^{-1}$.

## 2.2    Instrumentation

The data used in this study was collected during a summer campaign in and around Madrid in July 2016. Three air quality supersites were used, two urban stations and one suburban station, in addition to a setting in a suburban environment with two tethered balloons that allowed study of the vertical distribution of aerosol and air pollutants. All stations are located within a range of 17 km. A map displaying all locations is shown in Fig. S1.

The CSIC urban station, operative from 9 to 20 July, was located in the Institute of Agricultural Sciences (40°26'25" N,

03°41'17" W, 713 m a.s.l.) in central Madrid. The instrumentation in this station was installed in the sixth floor of the building. NO$_x$ and equivalent black carbon (BC) concentrations were measured with a chemiluminescence based analyzer (Teledyne API, 200EU) and an Aethalometer (AE33, Magee Scientific), respectively. The aerosol number size distribution was measured with a Scanning Mobility Particle Spectrometer (SMPS) (TSI, 3082) equipped with a nano-Differential Mobility Analyzer (DMA), for the size range 8-120 nm, and a Particle Size Magnifier (PSM) (AirModus) for the size range

1-4 nm.

The CIEMAT urban background station, operative from 4 to 20 July was located in Madrid, 4 km from the CSIC station (40°27'23" N, 03°43'32" W, 669 m a.s.l.). NO$_x$, O$_3$ and BC concentrations were measured with a chemiluminescence based analyzer (THERMO 17i), an ultraviolet photometry analyzer (THERMO 49i) and an Aethalometer (AE33), respectively. The aerosol number size distribution was measured with an SMPS (TSI 3080) for the size range 15-660 nm and a 1 nm

SMPS (TSI 3938E77) for the size range 1-30 nm. Temperature (4 m a.g.l.), relative humidity (4 m a.g.l.), solar radiation (35 m a.g.l.) and wind speed and wind direction (55 m a.g.l.) were measured at a meteorological tower at the station.

The ISCIII suburban station was located in the Institute of Health Carlos III, in Majadahonda, 15 km from the CSIC station (40°27'27" N, 03°51'54" W, 739 m a.s.l.) and was operative from 4 to 20 July. An SMPS (TSI 3080) measured the aerosol number size distribution in the size range 9-360 nm, and a PSM (AirModus) in the size range 1-4 nm. PTR-ToF-MS (Ionicon

Analytik, PTR-TOF 8000) operating in H3O+ mode was used to measure VOCs. Detailed description of the instrument of the instrument can be found in Graus et al. (2010). Operation procedure of the PTR-ToF-MS is fully described in Querol et al. (2017b).

Regarding the vertical measurements, two tethered balloons carrying miniaturized instrumentation were based at Majadahonda (MJDH) rugby field (40°28'29.9" N 3°52'54.6" W, 728 m a.s.l.), 17 km from CSIC. 28 flights up to 1200 m

a.g.l. were carried out from 11 to 14 July. A miniaturized SMPS (Hy-SMPS) measured the particle size distribution in the range 8-245 nm with a time resolution of 45 s and flow of 0.125 L/min (Lee et al., 2015). However, only particles larger than 10 nm could be detected due to a lower efficiency for finer particles. The instrument was inter-compared with the conventional SMPS (TSI 3080) installed at ISCIII station. The instruments were moderately correlated based on total particle concentration in the size range 9.14-241.44 nm: N$_{MJDH}$ = 1.3 N$_{ISCIII}$ - 124.5 cm$^{-3}$; R$^2$=0.47 (Fig. S2). Temperature,

relative humidity, pressure, wind speed and wind direction were also measured. The instrumentation was also equipped with



a Global Positioning System (GPS). An additional set of the miniaturized instrumentation was placed at surface level for comparison.

### 2.3 Data analysis techniques

Identification of NPF events was made by the method proposed by Dal Maso et al. (2005). After examination of the daily particle size distribution, if the day was classified as an event day we proceeded to calculate growth rates (GR), shrinking rates (SR), condensation and coagulation sinks (CS and CoagS) and formation rates ($J_{Dp}$).

The algorithm proposed by Hussein et al. (2005) was used to fit log-normal modes to the particle size distribution, from which GR were calculated following Eq. (1):

$$GR = \frac{dD_p}{dt},$$   (1)

where $D_p$ are the selected geometric mean diameters corresponding to growing particle modes. Unless stated otherwise, in this work growth rates are calculated for particles growing from 9 to 25 nm.  SR were calculated analogously when a decrease in the diameter of the fitted modes was observed.

CS, a measure of the removal rate of condensable vapor molecules due to their condensation onto the pre-existing particles
(Kulmala et al., 2012), is calculated using Eq. (2):

$$CS = 2\pi D \sum_i \beta_i D_{p,i} N_i ,$$   (2)

where D is the diffusion coefficient of the condensing vapor (here we use $H_2SO_4$), $D_{p,i}$ and $N_i$ are the particle diameter and particle concentration for the size class $i$. $\beta_i$ is the transitional correction factor

$$\beta_i = \frac{1+K_i}{1+\left(\frac{4}{3\alpha}+0.337\right)K_i+\frac{4}{3\alpha}K_i^2},$$   (3)

being K the Knudsen number $K_i = \frac{2\lambda}{D_{p,i}}$, where λ is the mean free path of the condensing vapor in air.

The formation rates of particles were calculated as a 30 minutes average, following Eq. (4):

$$J_{Dp} = \frac{dN_{Dp}}{dt} + CoagS_{Dp} \cdot N_{Dp} + \frac{GR}{\Delta D_p} N_{Dp},$$   (4)

where we use 9-25 nm for $N_{Dp}$, and CoagS is a quantification of the ability of the preexisting aerosol to scavenge newly formed particles. For its calculation we take the geometric mean diameter of the size ranges 9-25 nm. CoagS can be
calculated using Eq. (5):

$$CoagS_{Dp} = \sum_{D_{p'}} K\left(D_p, D_p'\right) N_{Dp'}$$   (5)

where K is the coagulation coefficient. For a detailed description of the parameters and their derivation, see Kulmala et al. (2012).

Additionally, bivariate polar plots of concentration have been used to relate wind speed and direction with total particle
concentration using PSM data by means of the R package *openair* (Carslaw and Ropkins, 2012).




## 3 Results and discussion

### 3.1 Meteorological context

Figure S3 shows the evolution of temperature, relative humidity, wind speed and wind direction measured at CIEMAT from 5 to 20 July 2016. The evolution of temperatures during this period evidences a succession of accumulation and venting episodes. Rain gauges collected significant precipitation only on 6 July at midnight (not shown).

The balloons field campaign, held from 11 to 14 July, coincided with the start of a venting period, coinciding with the passage of an upper level trough, and the transition to an accumulation period, when the trough has moved to the east of the Iberian Peninsula and a ridge passes over the area of study (see Fig. S4). Maxima and minima temperatures drop, while strong westerly winds predominate until they veer to NE on 12 July 18:00 UTC. High nocturnal wind speed peaks are recorded in this period, often accompanied by a change in wind direction. For detailed information on meteorological parameters during this campaign see Querol et al. (2017b).

### 3.2 Comparison of NPF events at urban and suburban surface stations

NPF episodes have been identified on a total of 12 days throughout the campaign. In Table 1 a summary of these events is presented. Out of these, a total of 6 days had simultaneous data available for at least one of the urban stations (CSIC, CIEMAT) and the suburban station (ISCIII). These episodes, marked with a star in Table 1, are selected for further analysis in this section. Figure 1 represents the aerosol particle size distribution of the selected episodes (12-18 July 2016).

#### 3.2.1 Episode characteristics

In the selected episodes, intensive daytime nucleation and subsequent condensational growth processes took place simultaneously at urban and suburban stations, located 17 km apart, and accordingly we classify these as regional NPF episodes. Nonetheless, some differences exist between urban and suburban events.

At urban stations particles of the order of 10 nm are detected throughout the day, even at night time. Conversely, at the suburban station such small particles are only detected during daytime. Additionally, during some days a very intense short nucleation burst is registered around midnight local time at urban stations that are not detected at the suburban station. This phenomenon will be analyzed later in this section.

Despite the detection of sub-10 nm particles as early as 04:00 UTC (06:00 local time) at the urban stations, only after around 09:00 UTC the growth of the particles is observed, occurring roughly at the same time in both urban and suburban stations. Newly formed particles grow until they have reached sizes of up to 50 nm, usually around 13:00 UTC (15:00 local time). After this, shrinkage is observed on 10 days, corresponding to 71 % of the days with available data. Consequently, the evolution of the particle size distribution is arc-shaped in these cases.

A further interesting feature is the presence of an additional Aitken mode on most days. Usually in the size range 50-100 nm, reaching 110 nm in some cases, this mode doesn't correspond to newly formed particles, but it follows a parallel evolution (condensational growth and potential shrinkage). When looking at the evolution of aerosol size distributions on consecutive days, it is possible to see a connection between this 50-100 nm mode and the distribution of the previous days. The nucleated and grown mode of one day is still present the following day and it continues to grow until it eventually fades away. In some occasions the Aitken mode can be tracked for two or more consecutive days, alternating stages of growth and shrinkage.





The start of the shrinking phase coincides with a marked increase in wind speed, therefore it is associated with dilution and possibly with a change in air masses. Figure 2 shows the shrinkage rates according to the starting diameter of the shrinking particles and the stations. Data used for this figure including start and end times and diameters is included in Table S1. The calculated shrinkage rates for particles with a starting diameter below 40 nm range from -1.1 to -8.0 nm h$^{-1}$. For particles in

the Aitken mode above 40 nm the values fall between -4.9 and -20.5 nm h$^{-1}$. Finally, particles that have reached the accumulation mode (i.e., particles greater than 100 nm) after their growth can shrink much faster, with values reaching -48.7 nm h$^{-1}$, the median being -12.7 nm h$^{-1}$. The results confirm that shrinkage is a regional phenomenon in the Madrid area, as already suggested by Alonso-Blanco et al. (2017), the processes being faster in the suburban station compared to the urban station. It is also observed that particles shrink faster the larger the starting diameter is.

It should be noted that nucleation episodes coincide in time with the early increases in O$_3$ concentrations in the morning, whereas the occurrence of maximum O$_3$ concentration (120 to 150 μg m$^{-3}$ hourly daily maxima between 14:00 and 16:00 UTC; see Fig. S5) takes place during the UFP growth stage, since oxidation of VOCs and inorganic gases is also accelerated with photochemistry and the presence of O$_3$ and OH radicals, among others (Coleman et al., 2008; Wang et al., 2017; Saiz-Lopez et al., 2017). NPF leads to maximal UFP concentrations around midday in all stations, which are recorded in

coincidence with very low BC levels.

### 3.2.2    Comparison of GR, J$_9$, CS and CoagS$_9$

For the observed daily regional NPF events, GR for the nucleation mode, $J_9$, CS and CoagS$_9$ have been determined using SMPS aerosol size distribution measurements. Figure 4 shows the growth rates according to urban and suburban stations.

Here, the growth rate is calculated from the time of detection of the smallest mode until either the particle reaches 25 nm or it stops growing before reaching that size. We considered only the events that are observed simultaneously at the suburban station and at least at one urban station. Growth rates were significantly higher at the suburban station, ranging 2.9-10.0 nm h$^{-1}$ with a median value of 5.8 nm h$^{-1}$, compared to values ranging 2.0-3.9 nm h$^{-1}$ with a median value of 2.7 nm h$^{-1}$ at the urban stations. This is consistent with the observed GR by Alonso-Blanco et al. (2017), ranging 1.4-10.6 nm h$^{-1}$ at CIEMAT.

The GR calculated in this study are also consistent with those observed in other urban and suburban areas. Kulmala et al. (2004) concluded that typical growth rates are 1-20 nm h$^{-1}$ in mid-latitudes. In particular, Stolzenburg et al. (2005) observed GR ranging 2.4-8.5 nm h$^{-1}$ in regional events in an urban environment in Atlanta, US. Qian et al. (2007) reported regional events with median GR of 5.1 nm h$^{-1}$ in an urban environment in St. Louis, US. Manninen et al. (2010) characterized NPF events in 12 European sites. Cabauw (The Netherlands) and San Pietro Capiofume (Italy), are stations located in

environments comparable to that in our suburban station, ISCIII. For these stations the observed median growth rates were 7-8 nm h$^{-1}$, corresponding well with our calculated GR in the suburban station.

Figure 5 shows the average daily cycles of total particle concentration in the size range 9-25 nm (N$_{9-25}$), CS and CoagS$_9$ during the regional NPF events at urban and suburban stations. N$_{9-25}$, average daily mean values are 3.7 x 10$^3$ cm$^{-3}$ and 2.2 x 10$^3$ cm$^{-3}$ at urban and suburban stations, respectively. The difference in number concentration in urban and suburban stations

could explain the difference in GR: if we consider that the vapor source is the same at all locations, it would be divided in bigger number of particles at the urban stations, thus giving slower GR. CS and CoagS$_9$ have average daily mean values of 3.4 x 10$^{-3}$ s$^{-1}$ (2.5 x 10$^{-3}$ s$^{-1}$ at the suburban station) and 2.4 x 10$^{-5}$ s$^{-1}$, respectively. After dawn, anthropogenic activities start, and N$_{9-25}$, CS and CoagS$_9$ start to increase at the same time, both in urban and suburban environments. Around 07:00 UTC, once the morning traffic rush diminishes, N$_{9-25}$ and the sinks increase more slowly; moreover, N$_{9-25}$ decreases in the suburban

station, indicating that in this environment the impact of the traffic emissions in total particle concentration is smaller than near the city center, as expected. Shortly after, at 09:00 UTC the photochemical processes are strong enough to start NPF, as





suggested by the increase in $N_{9-25}$, while the sinks get to a relative minimum. $N_{9-25}$ reaches its maximum at midday (9 x $10^3$ cm$^{-3}$ and 5 x $10^3$ cm$^{-3}$ at the urban and suburban stations, respectively), and then decrease because the particles start growing to diameters greater than 25 nm, adding to the sinks, which increase gradually until the evening. Around 19:00 UTC the effect of the afternoon traffic rush is evident, the variables evolving equivalently to that in the morning. Finally, at 23:00

UTC a sharp and short increase in $N_{9-25}$ is observed, associated with the aircraft emissions discussed above.

Total formation rates of 9-25 nm particles ($J_9$) were calculated from SMPS data of urban and suburban stations. Figure 6 summarizes the daily evolution of $J_9$ for the days in which an event is detected at both urban and suburban stations. Median hourly maximum values are registered at 10:00-12:00 UTC (2.0 cm$^{-3}$ s$^{-1}$) at the urban stations, and at 13:00-14:00 UTC (1.1 cm$^{-3}$ s$^{-1}$) at the suburban station. Median values are generally higher at the urban stations, especially during the central hours

of the day. The ranges are also broader at the urban stations, with ranges of up to 4 cm$^{-3}$ s$^{-1}$ at midday and outliers reaching 5 cm$^{-3}$ s$^{-1}$ at night, which agree with the observed nocturnal peaks at CSIC. The fact that $J_9$ is higher at the urban stations is probably linked to higher traffic emissions (20-30 nm, Brines et al., 2015) in the city, and not related with higher nucleation rates, since PSM measurements indicate lower concentrations of 1.2-4 nm particles compared to the suburban measurements (Figure S4). The calculated formation rates agree with those reported in other studies (see Kulmala et al. 2004 and references

therein), ranging 0.01-10 cm$^{-3}$ s$^{-1}$ during regional events around the world.

### 3.2.3    PTR-ToF-MS measurements

Among the 152 ions identified with the PTR-ToF-MS, only 3 exhibit temporal trends that might be relevant in the growth processes of NPF (Fig. S6). Two highly-oxygenated ions, $C_4H_4O_3H^+$ (m/z 101.023) and $C_2H_4O_3H^+$ (m/z 77.023), and $NO_2^+$

(m/z 45.9924) presented evolutions parallel to those of the particle diameter, i.e. the concentration of these ions increased simultaneously with the increase of particle diameter, and growth stopped when the concentration of the ions decreased (Fig. S6). This is observed also on days in which there is no particle formation but there is particle growth. Thus, the parent molecules of these ions are not linked to particle formation, but they would most probably contribute to particle growth. The fragment $C_4H_4O_3H^+$ has, to the best of our knowledge, only been reported once over an orange grove in California (Park et

al., 2013) and is most probably from secondary origin considering both its diurnal variation and oxidation state. It contains sufficient carbon atoms and oxygen functional groups to likely partition into the condensed phase. $NO_2^+$ and $C_2H_4O_3^+$ are known fragments of peroxyacetyl nitrate (PAN) (de Gouw et al., 2003), but $NO_2^+$ can also arise from the fragmentation of a wide range of peroxynitrates ($ROONO_2$) or alkyl and multifunctional nitrates ($RONO_2$) (Aoki et al., 2017; Duncianu et al., 2017). While the uptake of PAN on particles can be considered as negligible (Roberts, 2005), higher molecular weight

organonitrates are more likely to partition onto the particle phase. Thus, the particles growth appears to be driven by the uptake of secondary organic compounds. More precisely, in an urban atmosphere such as Madrid characterized by high $NO_2$ concentrations, the formation of organonitrates and/or peroxynitrates could play an important role in the particle growth processes.

### 3.2.4    Nocturnal UFP peaks


There are other interesting events taking place during night time. From 6 to 11 July and 17 to 19 July, high concentrations of 1.2-4 nm particles are registered shortly after sunset for several hours, simultaneously at urban and suburban stations (Fig. S7). BC, NO and $NO_2$ concentrations also increase during that time (see Fig. S5). Therefore, these processes are probably related to local traffic emissions and the decrease of the mixing layer after sunset. On the other hand, from 12 to 14 July high

concentrations of sub-25 nm particles are also detected, but only registered at the urban stations around 23:00 UTC. These



are sudden, shorter and more intense, with concentrations greater than $10^5$ cm$^{-3}$. They appear as intense bursts that last one hour or less, with no subsequent growth. Unlike the regional events, these are not accompanied by simultaneous high BC or NO concentrations, thus they are not linked to traffic emissions, although NO$_2$ levels are significant. Furthermore, these episodes occur outside local traffic rush hours, and are registered together with strong NE winds, which suggest that they

might be transported and not formed locally.

In order to determine the origin of these sub-25 nm particles, bivariate polar plots of concentration have been used to relate wind speed and direction measured at CIEMAT with total particle concentration of 1-4 nm particles, BC, NO$_2$ and NO measured at CSIC, separately analyzing daylight and night time periods (Fig. 3). These plots must be carefully interpreted, since the color scale only represents the average value for a given wind speed and direction. The results are consistent with

what we previously stated: the highest nocturnal 1-4 nm particle concentrations are linked with strong winds from NE direction. Air masses transported from this direction have the lowest BC levels, and moderate NO$_2$ concentrations. NO concentrations are insignificant at nighttime considering any direction, probably because of titration due to the high concentrations of O$_3$ observed during daytime. The airport Adolfo Suárez Madrid-Barajas is located NE of the city, 12 km from of the urban stations (see Fig. S1). With more than 34000 operations in July 2016 (AENA, 2016), it is the sixth busiest

airport in Europe. Other studies have linked aircraft emissions with nucleation bursts without growth (Cheung et al., 2011, Masiol et al., 2017), having the particle size distribution of the aircraft emissions in their area of study a characteristic mode peak at around 15 nm (Mazaheri et al., 2009). Considering that the airport of Madrid is relatively busy also at nighttime, and in regard of our results, we assume that the observed local nocturnal nucleation mode events are associated with transported aircraft emissions. Since the airport is located almost 30 km to the E of ISCIII, the emissions would be significantly diluted

by the time they arrive at that station, in comparison with the considered urban stations.

### 3.3    Vertical distribution of NPF events

#### 3.3.1    UFP concentrations

Querol et al. (2017b) studied the vertical profiles of UFP and O3 concentrations measured during the campaign using the balloon soundings at Majadahonda. UFP concentrations are homogeneous throughout the mixing layer and present a sharp decrease at the top. As the day progresses, the convection is more effective and high UFP levels reach higher altitudes, as the mixing layer heightens. This suggests that UFP fluxes are bottom-up. Moreover, the concentrations tend to increase until midday. Afterwards, they remain constant or slightly decrease, always showing homogeneous levels from surface levels to

the top of the PBL. Concentrations of UFP markedly increased from 11 to 14 July, both at surface and at upper levels. This is consistent with the observed decrease of the convective activity in that period, evidenced by a decrease in temperatures, but also with an increase in the formation rates, calculated in this study. Therefore, the increase in particle concentration is probably the result of both a decline in PBL height and more intense nucleation episodes.

#### 3.3.2    Particle size distribution and NPF episodes

The NPF events described in Sect. 3.2 that took place between 12 and 14 July were not only detected at surface level but also in upper layers with the balloons soundings in Majadahonda. However, the measurements were not continuous, since the balloons could not be operated safely if the wind speed was above 8 m s$^{-1}$ at any vertical level.





Figure 7 shows the fitted modes to the particle size distribution measured in the soundings on 12 July. The fact that sub-40 nm particles are not detected at the higher levels of the first flights suggests that convection is not very effective yet, and the sounding goes through different atmospheric layers, most likely the mixed layer and the residual layer. The interphase would be around 1300 m at 7:00 UTC, 1500 m at 9:00 UTC and higher than 1800 m from 10:00 UTC. From 10:00 UTC onwards, once the convection has fully developed, the mixing layer covers all the sounding and we see a homogeneous distribution at all levels, which is also comparable to those recorded with the instrumentation measuring at the surface. This agrees with the fact that UFP fluxes are bottom-up, as we stated in Sect. 3.3.1.

In the early morning the size distribution is dominated by a 60 nm mode at all altitudes, which grows to 100 nm at 11:00 UTC. Even though it is detected at all levels, the mode slightly decreases its size when the sounding ascends above the mixed layer limit, more clearly visible on the second flight, around 9:00 UTC. This result suggests that there are less vapors in the residual layer, which inhibits particle growth, whereas the mixed layer is more polluted, thus the particles can grow faster. The growth rates calculated for this mode were 1.8 nm h$^{-1}$ in the residual layer, and 7.3 nm h$^{-1}$ in the mixed layer. The concentration and size of the Aitken mode decrease after midday, which might be related to a change of air masses that entailed dilution and evaporation, leading to shrinking of the particles.

Moreover, during the morning we observed particles growing inside the mixing layer from 10 nm at 7:00 UTC, to 30 nm at midday, with a growth rate of 3.5 nm h$^{-1}$. The fact that the growth rate is the same throughout the mixing layer even though we expect VOCs to be higher near the surface upholds the assumption that the convection is very efficient, and the entire layer is well-mixed. After 13:00 UTC, because of the change in air masses particles start to shrink. While concentrations were not as high as other episodes, the evolution is remarkably similar to the NPF event measured at the same time at ISCIII, which had a growth rate of 3.0 nm h$^{-1}$.

The size distribution and the corresponding fitted modes for the soundings made on 13 July are presented in Fig. 8. Although the balloons could not fly until 10:30 UTC for safety reasons, 3 modes are detected from early morning at the sounding location. The accumulation mode grows from 156 nm at 07:00 UTC to 200 nm at 10:00 UTC, with a growth rate of 13.3 nm h$^{-1}$. After that time the particles have grown beyond the detection limit. This mode is also detected at ISCIII, growing 13.7 nm h$^{-1}$, following a similar evolution, but not at CSIC since this size range is beyond the detection limit of the SMPS. Another mode starting roughly at 40 nm at 09:00 UTC grows to 100 nm at 15:00 UTC. With a growth rate of 8.5 nm h$^{-1}$, this mode was detected at all altitudes once the soundings started, indicating that the convection was already effective by 10:30 UTC and all the measured altitudes were completely mixed, leading to a homogeneous particle distribution throughout the soundings. This mode is the prolongation of the Aitken mode detected the day before, which shrank from midday until the following morning. It is also detected at ISCIII and CSIC, with growth rates of 7.5 nm h$^{-1}$ and 6.9 nm h$^{-1}$. A nucleation mode grows from the detection limit of the instrument, around 10 nm at 08:30 UTC to 40 nm at 15:00 UTC. We consider this a regional NPF event, since the start of the particle growth is registered simultaneously at all the stations. The growth rates at the sounding location, ISCIII and CSIC are 5.3 nm h$^{-1}$, 4.6 nm h$^{-1}$ and 2.0 nm h$^{-1}$, respectively.

Finally, Fig. 9 shows the particle size distribution and fitted modes for the soundings made on 14 July. Correspondingly, in Fig. 10 the vertical distribution of particles for some of the soundings is presented. The earliest soundings revealed the existence of a residual layer aloft. In order to verify this result two constant altitude flights were made during the morning. The altitude was chosen so that the instruments remained initially outside the mixing layer, i.e. inside the residual layer. As the insolation increased, so did the altitude of the mixing layer, until it reached the altitude at which the balloons were positioned. As the mixing layer reached the balloons, total particle concentration sharply increased from $4 \times 10^3$ to $2 \times 10^4$ cm$^{-3}$, demonstrating that newly-formed particles flow upwards and remain inside the mixing layer.




According to the abrupt decline in particle concentration, the boundary between the mixing and residual layers was located at 1000 m at 09:00 UTC, 1200 m at 10:00 UTC, 1350 m at 11:00 UTC and beyond 1800 m after 12:00 UTC. This can be taken as an indicator of the effectiveness of convection, meaning that after 12:00 UTC all the measured particle population was well mixed throughout the sounding range. Inside the residual layer particles had a slower growth rate (0.5 nm h$^{-1}$

compared to 8.45 nm h$^{-1}$ for the 40 nm mode), and no particles smaller than 20 nm were observed.

The Aitken mode particles observed on the previous days prevailed and had grown to 170 nm at 08:00 UTC, reaching the detection limit (240 nm) by 10:00 UTC. Furthermore, nucleation mode particles were detected exclusively inside the mixing layer from 08:00 UTC to 12:00 UTC, whereas growth was only observed from 09:00 to 11:00 and from 12:00 onwards. The time spacing between both growing periods coincides with a marked decrease in wind speed. During the first period growth

rates at the sounding station, ISCIII and CSIC were 6.2, 5.4 and 1.4 nm h$^{-1}$, respectively. However, during the second stage particles grew faster at the urban station (8.6 nm h$^{-1}$) than at the sounding location (4.5 nm h$^{-1}$). As the latter is a suburban environment, this contrasts with the results obtained in Sect. 3.2.2. This fact could be explained by the veer of NE winds to weaker southerly winds in Madrid, which is not observed in Majadahonda.

Overall, the soundings revealed that there is simultaneous growth and shrinking of 3 different modes: nucleation, Aitken and

accumulation modes, and that all of them grow and shrink at very different rates. This was also observed in the surface measurements when comparing urban and suburban stations (see Sect. 3.2.2). This observation demonstrates that there are plenty of semivolatile vapors that condense onto and evaporate from the accumulation mode in this environment. Furthermore, the fact that the Aitken and nucleation modes have slower growth, implies that there is either less vapors that condense on the accumulation mode (they would need to be less volatile), or the smaller particles cannot uptake as much of

the vapors as the larger, due to particle phase chemistry (Apsokardu and Johnston, 2017). This phenomenon has been rarely reported in ambient air.

## 4      Conclusions

A total of 6 intensive regional NPF episodes were detected simultaneously at urban and suburban stations located within a 17

km radius in Madrid. The evolution of the size distribution was very similar at all stations: around sunrise nucleation mode particles appear and start growing and in the afternoon a decline in particle sizes, i.e. shrinkage, is observed. The shrinkage can be related to a change in air masses as suggested by the meteorological data. On most days one or more distinct modes were detected - in the Aitken or accumulation ranges - which can be tracked to previous and subsequent days. This implies that particles formed and grown one day can prevail for two or more days in the region if the meteorological conditions are

favorable. Some relevant differences between urban and suburban stations were observed. The urban stations presented larger formation rates and smaller growth rates as compared to the suburban stations. Additionally, a few nocturnal bursts of nucleation mode particles were observed in the urban stations, which could be related with aircraft emissions transported from the airport of Madrid. CS and CoagS evolutions were similar at all stations, having minimum values shortly before sunrise and increasing after dawn towards the maximum value after midday in the early afternoon. In general, the sinks were

higher at the urban stations. Formation rates were also larger in the urban stations, which might be related to traffic emissions that were considered in the size range used for the calculations.

We cannot determine whether the NPF episodes were triggered by the pollution generated in the city that extended to the region, or the events are caused by a broader phenomenon. In either way, it can be concluded that in summer the particle number concentrations are dominated by NPF in the area of study. The impact of traffic emissions on concentrations of UFP

is much smaller than those of NPF, even near the city center where the pollution load is higher. This result is in line with




other studies (e.g. Kulmala et al., 2016). Given the extent of the episodes, the health effects of NPF can affect a vast number of people, considering that the Madrid metropolitan area with more than 6 million inhabitants is the most populated area in Spain, and one of the most populated in Europe (UN, 2008). For this reason, we believe that the study of health effects related to newly-formed particle inhalation is crucial.

Regarding the vertical soundings of the NPF events, we observed that in the early morning the vertical distribution of newly formed particles is differentiated in two layers. The lower layer, in which convection is effective, is well-mixed and has a homogeneous particle size distribution. This layer heightens throughout the day, as insolation is more pronounced, extending beyond the sounding limits around midday. NPF occurs throughout this layer, and growth rates and concentrations are homogeneous. The upper layer is a stable residual one in which particles formed or transported the previous days prevail. In

the residual layer growth is inhibited or even completely restrained, compared with the same particles in the mixed layer. Overall, the soundings demonstrate that the flux of ultrafine particles has an upward direction and that particles are formed at surface levels, but they can prevail and be displaced and stored at upper levels and continue to evolve on following days.

Both at surface and aloft we detected growth and shrinkage of the nucleation, Aitken and accumulation modes, which had very different growth and shrinking rates. This suggests that there are vapors that can condense onto and evaporate from the

accumulation mode, which has been rarely documented to this day.

In this campaign we could not measure in the earliest stages of NPF due to safety requirements of the balloon flights early in the morning. We think it is important for future work to carry out soundings during the nucleation phase of the episodes. However, miniaturized instruments with greater resolution would be needed, which are not available at the present time. This would allow us to determine whether particle formation takes place throughout the mixing layer or occurs at the surface and

is transported upwards by convection afterwards. If the former were true, then locations with high PBL could produce more particles than we have considered, and they could affect larger populations if they were transported to surface levels, or affect the climate to a greater extent, since newly-formed particles can be activated as cloud condensation nuclei once they grow beyond 50 nm, thus affecting the radiative balance of the Earth.

**Acknowledgments**

This work was supported by the Spanish Ministry of Agriculture, Fishing, Food and Environment, the Ministry of Economy, Industry and Competitiveness, the Madrid City Council and Regional Government, FEDER funds under the project HOUSE (CGL2016-78594-R), the CUD of Zaragoza (project CUD 2016-05), the Generalitat de Catalunya (AGAUR 2017 SGR44) and the Korean Ministry of Environment through "The Eco-Innovation project". The funding received by ERA-PLANET

(www.era-planet.eu), trans-national project SMURBS (www.smurbs.eu) (Grant agreement No. 689443), and support of Academy of Finland via center of excellence in Atmospheric sciences are acknowledged. These results are part of a project (ATM-GTP/ERC) that has received funding from the European Research Council (ERC) under the European Union's Horizon 2020 research and innovation program (Grant agreement No. 742206). Authors also acknowledge the Doctoral program of Atmospheric Sciences at the University of Helsinki (ATM-DP). M.K. acknowledges the support by the Academy

of Finland via his Academy Professorship (no. 302958). We also thank the City Council of Majadahonda for logistic assistance, and Instituto de Ciencias Agrarias, Instituto de Salud Carlos III, Alava Ingenieros, TSI, Solma Environmental Solutions, and Airmodus for their support.

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



Table 1: Summary of new particle formation events recorded during the campaign showing the starting time, considered as the moment of first detection of the nucleation mode, the final time, considered when the mode reaches 25 nm, and the growth rate calculated in that period. A star marks the events that are detected simultaneously at all stations and were chosen for further analysis in this work.

| | Date | Starting Time (UTC) | Final Time (UTC) | GR (nm h$^{-1}$) |
|---|---|---|---|---|
| CSIC | 12/07/2016 (*) | 6:20 | 10:39 | 3.9 |
| | 13/07/2016 (*) | 8:30 | 12:49 | 2.0 |
| | 14/07/2016 (*) | 8:45 | 11:53 | 1.4 |
| ISCIII | 12/07/2016 (*) | 5:30 | 9:44 | 3.0 |
| | 13/07/2016 (*) | 8:50 | 11:54 | 4.6 |
| | 14/07/2016 (*) | 9:20 | 10:39 | 5.4 |
| | 16/07/2016 (*) | 11:30 | 13:14 | 10.0 |
| | 17/07/2016 (*) | 8:25 | 10:20 | 7.1 |
| | 18/07/2016 (*) | 10:44 | 12:20 | 2.9 |
| CIEMAT | 15/07/2016 | 8:34 | 13:08 | 4.0 |
| | 16/07/2016 (*) | 9:40 | 13:03 | 2.8 |
| | 17/07/2016 (*) | 9:55 | 13:00 | 2.8 |
| | 18/07/2016 (*) | 9:09 | 11:49 | 2.6 |
| MJDH -Sounding | 12/07/2016 | 7:27 | 8:08 | 3.5 |
| | 13/07/2016 | 8:39 | 9:56 | 5.3 |
| | 14/07/2016 | 9:00 | 10:34 | 6.2 |

**Figure captions**

Figure 1: Particle size distribution of the regional new particle formation episodes detected during the intensive field campaign (a) at CSIC (up) and ISCIII (down) in 12-14 July, and (b) at CIEMAT (up) and ISCIII (down) in 16-18 July. For CIEMAT data only the long SMPS is represented here.

Figure 2: Boxplot of shrinkage rate (SR) determined during regional NPF events at urban (CSIC and CIEMAT) and suburban (ISCIII) stations according to the starting diameter of the shrinking particles. The considered categories are: particles with a starting diameter below 40 nm, particles with a starting diameter between 40 and 100 nm, and particles with a starting diameter larger than 100 nm (Acc). The red line represents the median, the upper and lower limits of the boxes represent the 75th and 25th percentiles and the whiskers include 99.3% of the data. Outliers are represented with a red cross.

Figure 3: Bipolar plot of (a) total particle concentration in the size range 1.2-4 nm measured with the PSM, (b) Black Carbon, (c) $NO_2$ and (d) NO concentrations at CSIC urban station, using the wind data registered at CIEMAT. Daylight and nighttime hours are separated according to sunrise (5 UTC) and sunset (20 UTC) hours. The data correspond to the period 11-15 July 2016.

Figure 4: Boxplot of growth rates (GR) determined for the nucleation mode (< 25 nm) during regional new particle formation events at urban (CSIC or CIEMAT) and suburban (ISCIII) stations. The red line represents the median, the upper and lower limits of the boxes represent the 75th and 25th percentiles and the whiskers include 99.3% of the data. Outliers are represented with a red cross.

Figure 5: Averaged daily cycles of (a) total particle concentration in the size range 9-25 nm, (b) Condensation Sink (CS) and (c) Coagulation Sink ($CoagS_9$) during regional new particle formation events at urban (CSIC and CIEMAT, solid line) and suburban (ISCIII, dashed line) stations. The hour of the day is UTC. Local time is UTC+2.

Figure 6: Hourly formation rates in the ranges 9.1-25 nm at (a) urban (CSIC or CIEMAT) and (b) suburban (ISCIII) stations during regional new particle formation events. The red line represents the median, the upper and lower limits of the boxes represent the 75th and 25th percentiles and the whiskers represent the 5th and 95th percentiles. Time is UTC. Local time is UTC+2.

Figure 7: Particle size distribution with fitted log-normal modes measured during the balloons soundings at Majadahonda on 12 July 2016. The altitude of the instrumentation is represented with a white line. Surface level is 630 m above sea level. Time is UTC. Local time is UTC+2.





Figure 8: Particle size distribution with fitted log-normal modes measured during the balloons soundings at Majadahonda on 13 July 2016. The altitude of the instrumentation is represented with a white line. Surface level is 630 m above sea level. Time is UTC. Local time is UTC+2.

5   Figure 9: Particle size distribution with fitted log-normal modes measured during the balloons soundings at Majadahonda on 14 July 2016. The altitude of the instrumentation is represented with a white line. Surface level is 630 m above sea level. Time is UTC. Local time is UTC+2.

Figure 10: Vertical particle size distribution measured on 14 July during selected soundings.

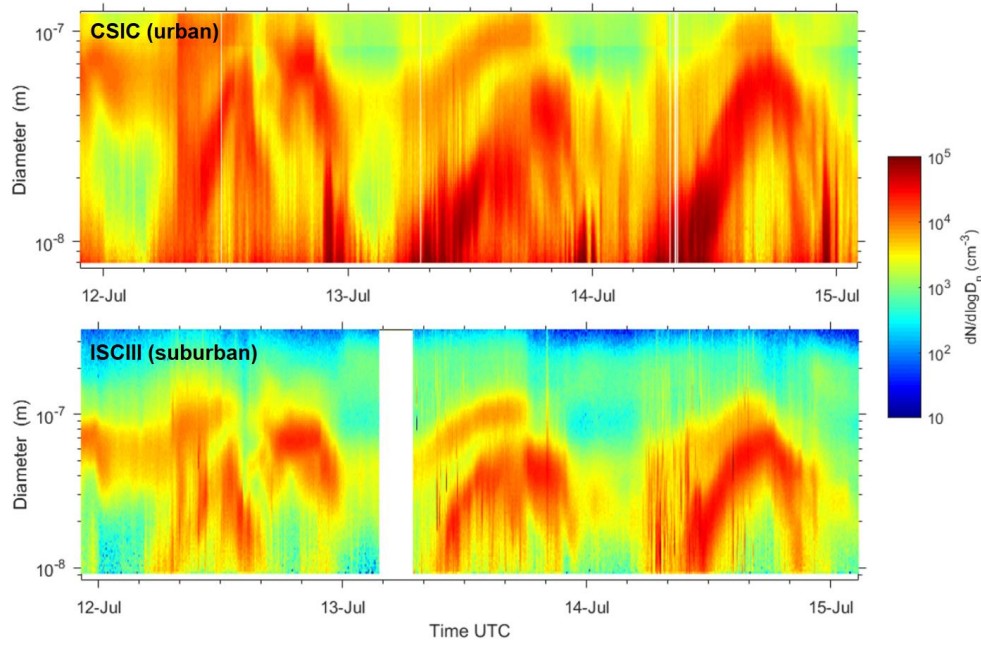

Figure 1a





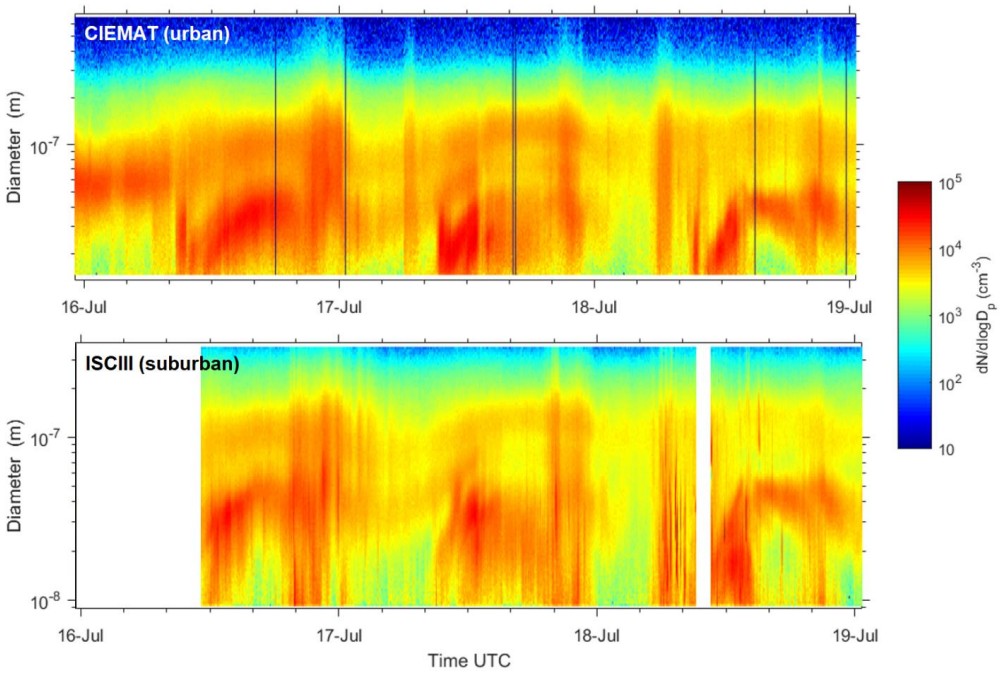

Figure 1b

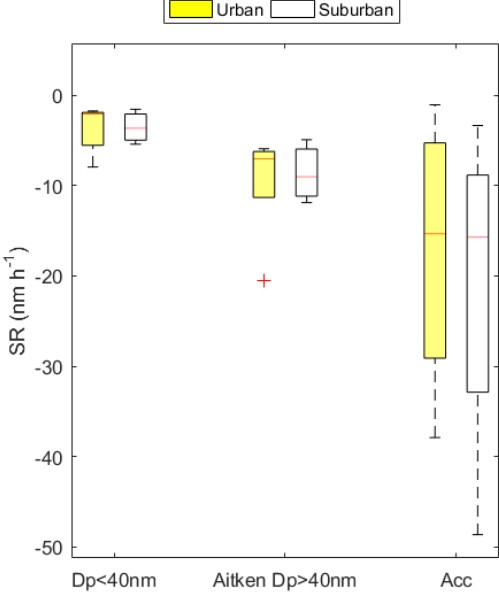

Figure 2

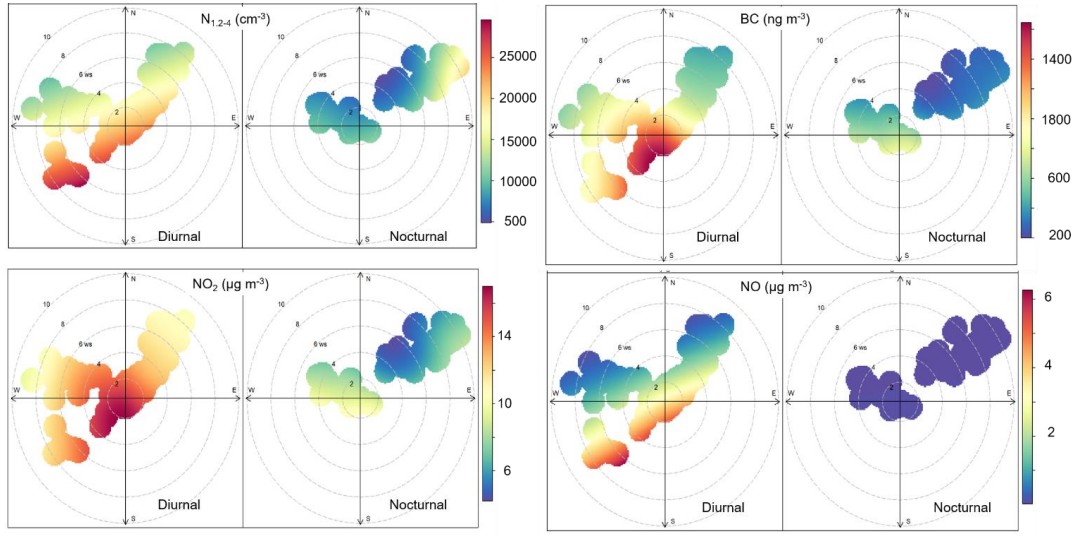

Figure 3

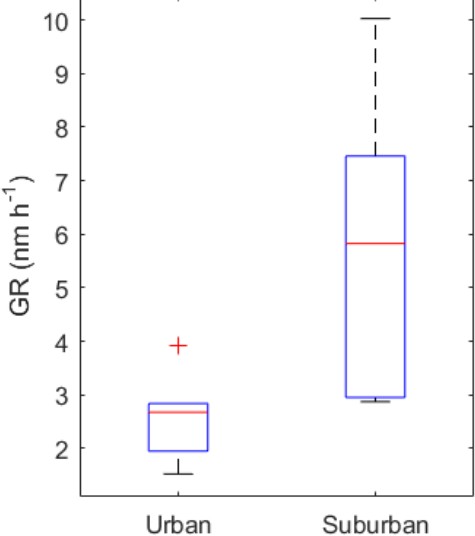

Figure 4





Figure 5

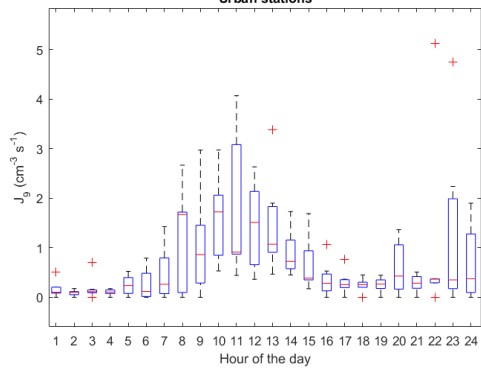

Figure 6a





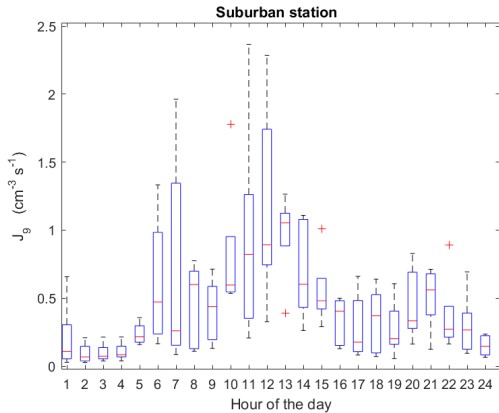

Figure 6b

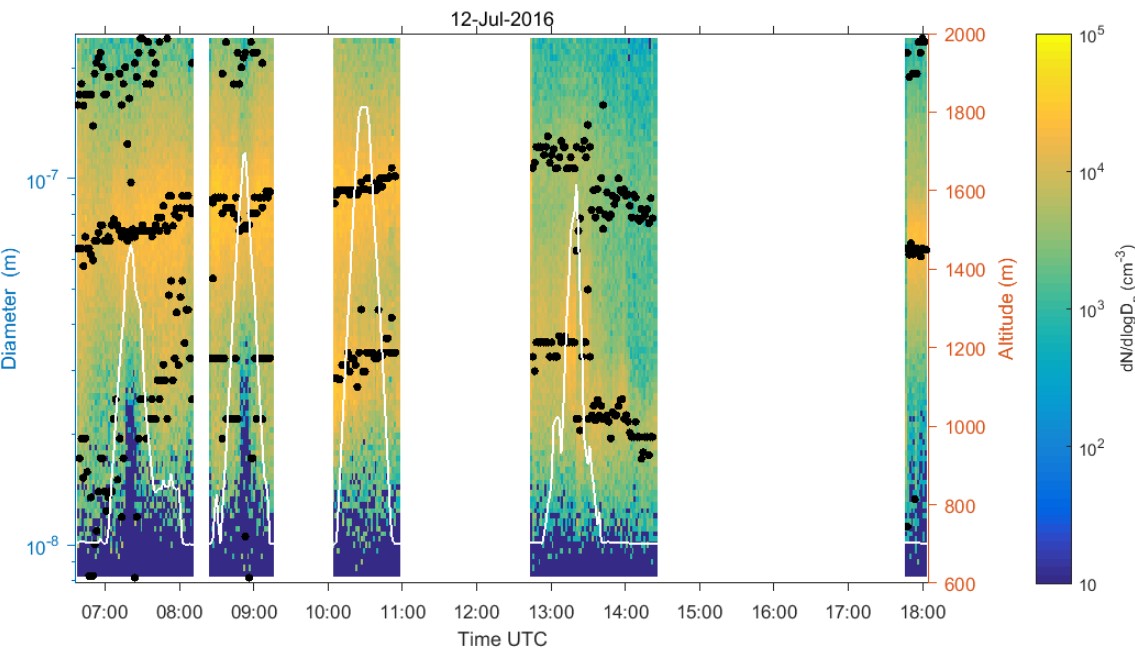

Figure 7



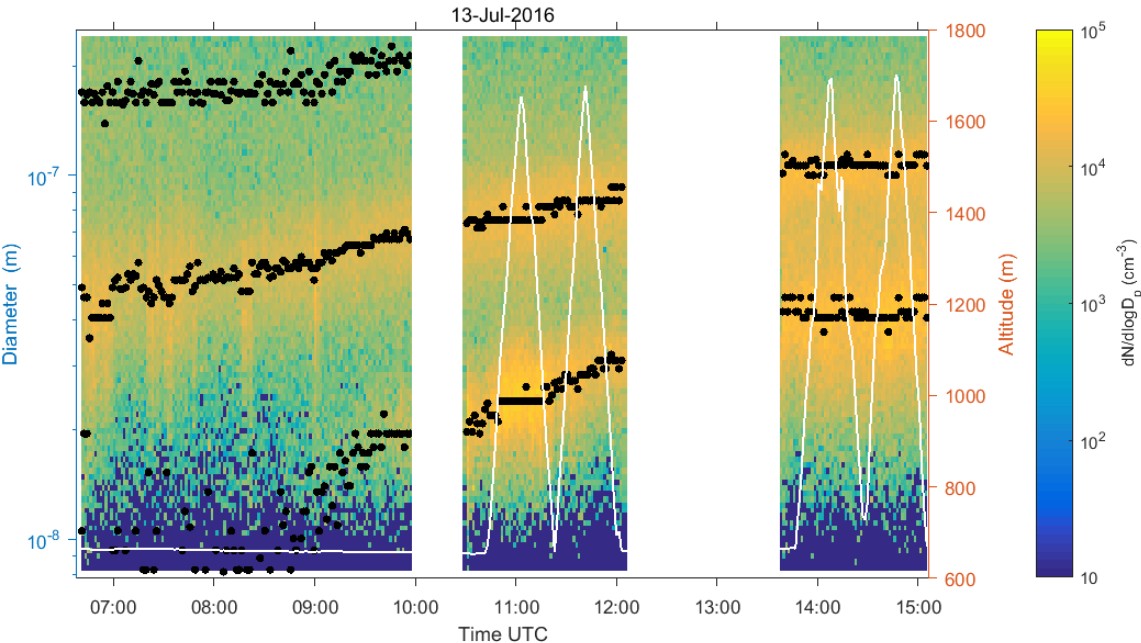

Figure 8

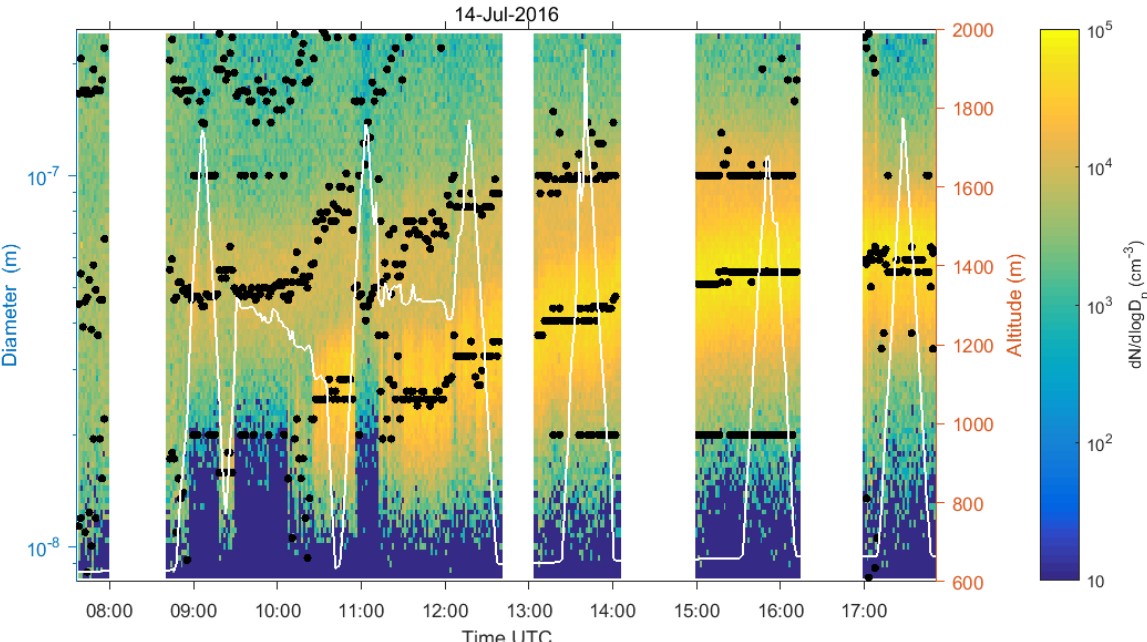

Figure 9



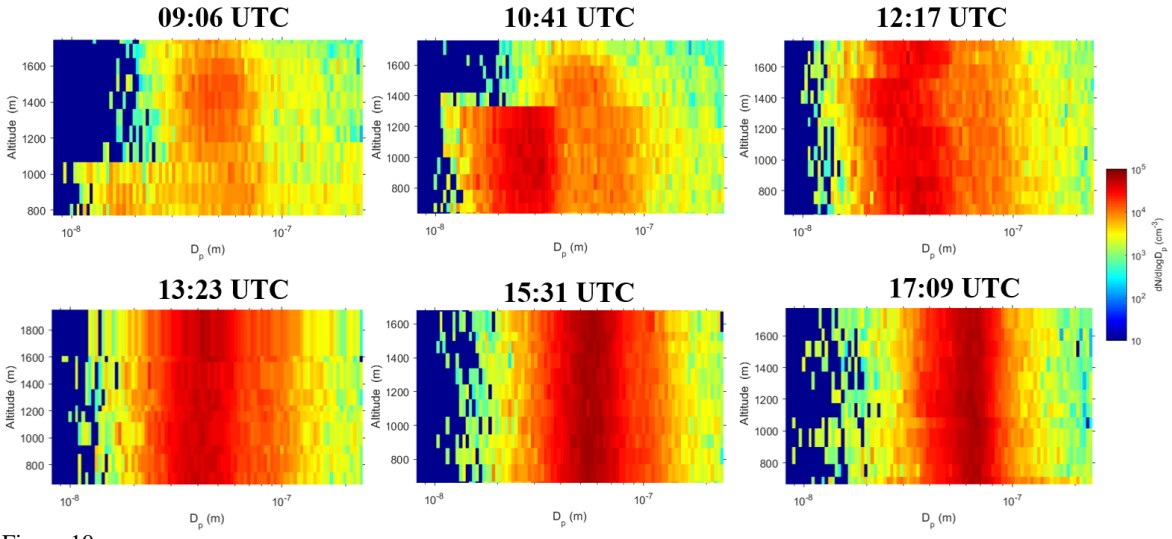

Figure 10