# Peer review of "Vertical and horizontal distribution of regional new particle"

_Atmospheric Chemistry and Physics, 2018_

## Referee Comment (RC1) · Anonymous Referee #1 · 18 Apr 2018

In this study, new particle formation (NPF) in Madrid during July 2016 is investigated from ground-based measurements of aerosol particle size distributions at three different locations in the Madrid area combined with vertical profiles at one location. The authors frequently observe ultrafine (UF) particles appearing in the morning followed by growth which they define as NPF events. On several days, such NPF events are observed simultaneously at the three different locations which leads to the conclusion that the NPF events are regional. From the analysis of the vertical profiles, the authors conclude that the UF particles are produced close to the surface and transported to higher altitudes within the mixed layer. New particle formation in urban locations have been investigated rather intensively in previous years. However, as the authors state in the introduction, rather few of the previous studies on urban NPF have included measurements in the vertical dimension. Therefore, I think the topic of this manuscript is suitable for publication in ACP. The manuscript is well structured and mostly well written. However, there are a few issues that the authors need to clarify, and possibly reassess, before publication.

Specific comments

1.) My first comment is related to the definition of primary aerosol particles in an urban location. A quite large part of the Introduction is spent on what fractions have been primary particles and originating from NPF in earlier studies in urban environments. Furthermore, since you claim that you can distinguish NPF events from primary emissions in this study, I think it should be more clear exactly how you define these two processes. In the Introduction, in lines 18-20 on page 2, you discuss production mechanisms of ultrafine particles from traffic: "condensation of semi-volatile phases vapor species that creates new UFP during dilution and cooling of engine exhaust emissions near the source". After that you write "most studies consider them as primary", "or quasi-primary particles". How do you define these particles, that form by nucleation in the tailpipe or a second after exiting? I guess you define them as primary, but I think that should be clear.

2.) Comment 1) leads to the question how you know that the regional NPF events are not "primary aerosol particles" formed by nucleation in the tailpipe or soon after exiting. Such emissions of 6-11 nm particles (Kittelson et al., 2006) dominate the number emissions. Furthermore, such emissions from thousands of cars in the Madrid area would likely look like a regional NPF event, since the emitted particles will grow by condensation in the atmosphere. One argument against this hypothesis, that the particles are primary, is the fact that your formation rates and number concentrations of 9-25 nm particles peak at noon when BC levels are at minimum. On the other hand, condensational growth is strongest at noon or in the afternoon since photochemical production of condensable vapors is dependent on solar radiation. Therefore, even though primary emissions are likely highest in the morning and evening rush hours,

the likeliness that the emitted nano-particles grow into the 9-25 nm interval (which you refer to as ultrafine particles and for which you present the diurnal cycle in Fig. 5a) may be highest at noon or in the afternoon. I think you need to strengthen your arguments and definitions here when you refer to the events as "regional NPF events". Also, could you add diurnal cycles of number concentrations associated with the PSM data to Fig. 5a?

3.) Following up on Comment 2), in Sect. 3.2.1 you equate the occurrence of "10 nm particles" with NPF. Could these particles not just as well be primary?

4.) Page 5, lines 22-23: Why did you choose the number concentration in the 9-25 nm interval in Eq. 4 for your definition of the formation rate? Why do you not use your PSM data for the formation rates?

5.) I do not follow the conclusion written in the abstract on lines 37-39: "The vertical soundings demonstrated that ultrafine particles (UFP) are transported from surface levels to higher levels, thus newly formed particles ascend from surface to the top of the mixing layer", or that the fluxes are "bottom-up" as written on page 9, line 28. As far as I can tell, Fig. 10 (which is a very nice figure by the way) only shows that the particles are produced inside the mixed layer (not the residual layer). The mixed layer of course grows during the day, but how can you tell whether the particles are being produced close to the surface or at the top of the mixed layer (or both)?

6.) Regarding shrinkage in lines 1-2 on page 7: Could evaporation be a reason for the shrinkage as well?

7.) Page 7, lines 30-31: "For these stations the observed median growth rates were 7- 8 nm h-1". These values are very close to the average growth rate of 7.3 nm h-1 reported from Bakersfield, a polluted location in California (Ahlm et al., 2012). Please also add this reference to the studies of NPF events in urban environments in lines 25-30 on page 2.

8.) Page 9, line 8. It seems this is the first time you discuss Fig. 3 so perhaps you should change the order of the figures.

9.) I suppose the black dots in Figs. 7-9 represent the fitted log-normal modes, but please add that information to the figure captions.

10.) You don't draw any conclusions from Fig. 9 so perhaps that figure is not necessary.

References

Ahlm, L., Liu, S., Day, D. A., Russell, L. M., Weber, R., Gentner, D. R., Goldstein, A. H., DiGangi, J. P., Henry, S. B., Keutsch, F. N., VandenBoer, T. C., Markovic, M. Z., Murphy, J. G., Ren, X., and Scott, S.: Formation and growth of ultrafine particles from secondary sources in Bakersfield, California, J. Geophys. Res. Atmos., 117, D00V08, doi:10.1029/2011JD017144, 2012.

Kittelson, D.B., Watts, W.F., Johnson, J.P.: On-road and laboratory evaluation of combustion aerosols – Part1: Summary of diesel engine results. J. Aerosol Sci. 37, 913-930, 2006.

---

## Referee Comment (RC2) · Anonymous Referee #2 · 24 Apr 2018

The paper *Vertical and horizontal distribution of regional new particle formation events in Madrid* from C. Carnerero *et al.* presents an interesting dataset about new particle formation in the metropolitan area of Madrid.

A proper understanding of NPF in urban environments is still missing despite the potentially high impact that ultrafine particles could have on the total particle number concentration and their health effects. To address this issue field and laboratory experiments are needed and the results presented in this work provide new insights in this direction. By providing a horizontal and vertical distribution of new particle formation episodes, the authors clearly show that these events are occurring over a large part of the metropolitan area underlining their impact on a large region. In my opinion this paper falls within the scope of ACP but there are several issues that need to be addressed before publication.

**Major comments**

- Despite the interesting dataset this paper fails in conveying a clear message due to its structure and the way the results are presented. In particular the authors mix the main results (horizontal and vertical characterization of NPF in Madrid) with episodes (e.g. the UFP peaks at night) and phenomena (e.g. shrinkage of particle size) that are not strictly related with them. To address this issue I suggest to the authors to focus more on the important results and emphasize them in a clearer way as well as to restructure the results section in order to make a clear separation between these results and all the minor observations.

- The authors state several times that NPF is dominating the total particle concentration in Madrid, however it is not clear how they separate between newly formed particles and ultra fine particles directly emitted from cars. I assume that the formation rate calculation is biased by the fact that primary UFP were not taken into consideration and this would explain why formation rates at the urban stations are higher than those measured in the suburban whereas the growth rates are smaller. The authors need to quantitatively estimate the source of UFP and revise the formation rates excluding primary UFP from their results. Doing this would also allow to compare directly the Madrid case with the other locations reported in the introduction. Probably the simplest way to discriminate between primary and secondary UFP would be to use the data of the particle concentrations below 9 nm, that should be available at all the measurement sites.

- In section 2.2 no details are provided about the sampling conditions. The authors should give a short description of the inlets and explain for example if losses were measured and/or calculated, if any intercomparison between the different particle counters was performed, etc... Moreover, a big part of the work is based on the measurements performed with the Hy-SMPS but no proper characterization is provided (figure S2 is not really useful to evaluate the performances of this instrument) and the only cited paper is written in Korean (I'm also not sure whether this is a peer reviewed journal or not). For these reasons a more complete characterization of this instrument is required, for example it could be useful to compare the Hy-SMPS with a reference SMPS while looking at separate size bins and not only at the total concentration.

- In section 2.3 it is explained how formation and growth rates are calculated, however no explanation is provided about the decision of using the 9-25 nm range, despite the fact that measurements of particle concentrations down to about 1nm were performed. I would argue that this is not the best choice, in particular for the formation rates that could be highly biased by primary emissions as previously explained. Moreover, using a smaller reference diameter for the formation rate would permit to better compare with measurements performed in other locations and/or in chamber studies. For these reasons I would suggest to calculate formation rates for a more meaningful size (below 5 nm) and eventually to calculate growth rates at two different sizes (for example keep the 9-25 nm range for comparability with the vertical sounding and add a lower size GR depending on the availability of the existing data). Finally, uncertainties for all the growth and formation rates should be estimated in order to make a proper comparison between different locations and days.

- Section 3.2.2 reports a comparison of growth rates and formation rates at the different sites but without including the vertical profiles (in this case growth rates are provided in section 3.3). I think that the paper would benefit by having all the growth rates presented together, this would improve the readability and the clarity of the paper.

- Section 3.2.3 reports PTR measurements of 3 ions that show some correlation with the particle growth, however this section is not adding any valuable information to the overall picture of the paper. For this reason I would suggest to either remove it or expand it with more detailed analysis. For example it would be worth investigating the possible precursors for HOMs formation. It may be possible to say something about the origin of the condensable vapours by looking at the concentration and diurnal profiles of biogenic vs. anthropogenic

VOCs.

- Section 3.2.4 should be written in a more consistent way, in particular the authors first speak of sub-25 nm particles but then they only shows bivariate plots of sub-4nm particle concentration, what is the reason for this? Moreover to better support the *airport hypothesis* it would be useful to show a time series of UFP for the period of interest together with wind speed and wind direction (extrapolating these data from the supplementary is difficult due to the long time period reported there). I'm asking this because the bivariate plots only show that UFP particle concentration is higher when the wind is coming from NE but to support the authors hypothesis it would be important to check if there are periods with low UFP concentration under the same wind conditions. If this is the case then I wound find the hypothesis less convincing due to the fact that the airport should be a more or less stationary source of UFP.

- In section 3.3 the authors often speak of "bottom-up flux" for the UFP however, what vertical profiles show is only that the UFP concentration is homogeneous inside the mixing layer. For this reason one should avoid using this terminology and the authors should correct all the corresponding parts in the paper. Moreover, I found that the interpretation of the vertical profiles graphs is complicated by the absence of a direct measurement of the mixing layer height. Referring to the "twin paper" [1], this information should be available for all the soundings (for example the potential temperature or total particle concentration can be used) and it would be a really nice addition to the graphs.

- The conclusions are written as a summary of the paper but here the author should focus more on the significance of their findings compared with existing observations. This section should be rewritten in order to convey a clearer message, so I suggest to the authors to delete all the unnecessary parts focusing more on the important results of the paper.

**Minor comments**

- Page2 line 3: "The NPF events extend over the full vertical extension of the mixed layer reaching as high as 3000 m." But the maximum height of the sounding is 2000 m, so this should be corrected.

- Page 2 line 4: "This can have consequences in the radiative balance of the atmosphere and affect the climate", the climatological effect of NPF in a polluted environment as Madrid is questionable and not

supported by any evidence in this paper, for this reason remove or rephrase this sentence.

- Page 2 line 5: As previously stated, a proper estimation of NPF over primary UFP should be given here.

- Page 2 line 25: There is no need to cite 25 papers, this can be reduced.

- Page 3 line 13: The sentence seems to contradict the cited paper of Querol et al.[1] where it is said that "Relatively low concentrations of ultrafine particles (UFPs) were found during the study, and nucleation episodes were only detected in the boundary layer."

- Page 4 line 13: It would be useful to add a scale to the map in figure S1.

- Page 4 line 19: It is not clear if the PSM was operated in scanning mode or fixed mode, it would be good to specify this.

- Page 6 line 14: It is said that NPF was identified on 12 days but table 1 only reports 7 days so this should be made consistent. Moreover, figure S8 shows that there were 2 nucleation events at CIEMAT on the 13th and 14th of July that are not mentioned in the table. In addition, it would be useful to add also formation rates to the table together with the GR values.

- Page 6 line 27: It's almost impossible to see the early morning UFP concentration just by looking at the full time series. Thus a dedicated plot should be made either by plotting sub-10nm particles on top of figure 1 or by plotting a diurnal profile for this size range.

- Page 7 line 1: Particle size shrinking is an interesting phenomenon but it doesn't fit nicely in this part of the text. For this reason describe it in a separate (small) section. It would also be nice to plot the wind speed specifically for the shrinking phase because from the overall plot it is difficult to see a trend. In some cases also a rapid change in number concentrations is observed pointing to a change in air mass. In these cases it does not make sense to speak about a shrinking of aerosols because the aerosol population changes. The authors should demonstrate clearly, that the same population of aerosols is shrinking. Otherwise, it is not shrinking and they should delete this.

- Page 7 line 14: Here a reference to figure 5 would be useful. Moreover, I think that associating UFP with particles in the range 9-25 nm is misleading and it would be better to plot the diurnal profile for all particles below 25 nm and for the total particle concentration.

- Page 8 line 5: "above" should be replaced by "below".

- Page 8 line 11: "The fact that J9 is higher at the urban stations is probably linked to higher traffic emissions [...] in the city, and not related with higher nucleation rates, since PSM measurements indicate lower concentrations of 1.2-4 nm particles". Here the authors confirm my hypothesis that primary particles affect formation rates, underlining the necessity to take this process into account in their calculation.

- Page 8 line 14: This is not figure S4 but figure S7.

- Page 8 line 14: "The calculated formation rates agree with those reported in other studies, ranging 0.01-10 $cm^{-3}s^{-1}$ during regional events around the world." I don't see any reason for reporting an agreement within 4 orders of magnitudes. The formation rates measured during this campaign should be compared in a more targeted way with other locations around the world.

- Page 8 line 30: "Thus, the particles growth appears to be driven by the uptake of secondary organic compounds." This is a reasonable assumption but cannot be proven by the PTR measurements presented in this work. Try to support this assumption with additional information, for example one can try to check if the growth rates can be explained by sulfuric acid alone (assuming a reasonable range of values for sulfuric acid concentration) or not.[2, 3]

- Page 10 line 10: "the mode slightly decreases its size when the sounding ascends above the mixed layer limit", as already written above the mixing layer height should be plotted together with the particle size distribution to better visualize these changes.

- Page 10 line 12: It would be good if the authors could specify how they calculated the growth rate in the residual layer. The impressions from the graphs is that there are really few points inside the residual layer and it is not clear whether there is any growth at all inside this layer.

- Page 10 line 15: I'm not really convinced by the presence of a 10 nm mode in the first sounding, Maybe there is an over-fitting issue with the mode fitting algorithm. For this reason I think calculation of the growth rate is questionable and should be avoided.

- Page 10 line 23: "The accumulation mode grows from 156 nm at 07:00 UTC to 200 nm at 10:00 UTC", also in this case I think the accumulation mode is over-fitted, the authors should either revise their

fitting algorithm or prove that I'm wrong by reporting in the SI a single SMPS scan plot with the fitted modes on top of it.

- Page10 line 26: "Another mode starting roughly at 40 nm at 09:00 UTC" I guess this should be 07:00 UTC.

- Page 10 line 30: I don't see any nucleation mode earlier than 09:30-10.00 UTC. Correct this sentence and eventually revise the calculated growth rate.

- Page 10 line 38: "As the insolation increased, so did the altitude of the mixing layer, until it reached the altitude at which the balloons were positioned." By looking at the plot it seems more likely that the balloon height decreased until reaching the mixing layer.

- Page 10 line 40: As previously explained avoid speaking of particles flowing upward, measurements are just showing that UFP are homogeneous inside the mixed layer.

- Page 11 line 5: I do not see a growth of 40 nm particles in the residual layer. The size of this mode is the same at the 9 and 11 UTC sounding.

- Page 11 line 6: how do you know that you observed these particles already the previous day? Moreover, also here I think that the Aitken mode is over-fitted.

- Page 11 line 16 and following lines: I would avoid speaking of the accumulation mode. I'm really sceptical about the presence of this mode in the measurements presented here and, even if it is present, then it is above the detection limit for most of the time. Moreover, it is said that the accumulation mode grew faster than the other modes and "this phenomenon has been rarely reported in ambient air." I think the data do not support this conclusion. If I'm not mistaken the fitted accumulation mode shows a growth only for a couple of hours on a specific day and the data are quite scattered so it doesn't seem like the growth is significantly higher compared with the Aitken mode. If the authors want to support this observation then they should try to look if anything similar is present in the SMPS ground measurements.

- Page 12 line 13,14: As already explained the vertical profiles do not show a clear accumulation mode and this is particularly true for the residual layer, so I would remove this sentence.

- Page 12 line 18: the authors don't need a miniaturized instrument with "greater resolution" but an instrument able to measure smaller particles.

- Figure 1: I would greatly recommend to avoid using jet colormap (*i.e.* rainbow colormap) for surface plots. This colormap is not perceptually uniform and this can create several kinds of issue as widely documented elsewhere (e.g. https://www.ncbi.nlm.nih.gov/pubmed/22034369 and http://ieeexplore.ieee.org/document/4118486/?reload=true)

- In figure S8 the authors report the total size distribution for the three measurement sites. This is a useful supplementary information but the readability of the graph should be improved. In particular a logarithmic color scale should be used as well as a higher image resolution. I also suggest to extrapolate the total particle number concentration for the 3 sites in the same size bins for better comparability rather than using different sizes for each site. Finally I noticed that there are some mismatches in the merged size distributions measured at CIEMAT (*e.g.* 8/7/2016). This would indicate that one of the instruments was not working properly. Please comment on this. How would this affect the presented results?

**References**

[1] X. Querol et al. "Phenomenology of summer ozone episodes over the Madrid Metropolitan Area, central Spain". In: *Atmospheric Chemistry and Physics Discussions* 2017 (2017), pp. 1–38. DOI: 10.5194/acp-2017-1014. URL: https://www.atmos-chem-phys-discuss.net/acp-2017-1014/ (cit. on pp. 3, 4).

[2] T. Nieminen et al. "Sub-10 nm particle growth by vapor condensation-effects of vapor molecule size and particle thermal speed". In: *Atmospheric Chemistry and Physics* 10.20 (2010), pp. 9773–9779. ISSN: 16807316. DOI: 10.5194/acp-10-9773-2010 (cit. on p. 5).

[3] Jasmin Tröstl et al. "The role of low-volatility organic compounds in initial particle growth in the atmosphere". In: *Nature* 533.7604 (2016), pp. 527–531. ISSN: 0028-0836. DOI: 10.1038/nature18271. URL: http://www.nature.com/doifinder/10.1038/nature18271 (cit. on p. 5).

---

## Author Comment (AC1) · 6 Jun 2018

**REPLY TO #1 REFEREE'S QUERIES AND DESCRIPTION OF CHANGES DONE FOLLOWING HER/HIS SUGGESTIONS**

Many thanks for your valuable comments and suggestions than helped us to improve the quality of the manuscript. As you will see we took into account all your comments when reviewing the new version.

1.) **My first comment is related to the definition of primary aerosol particles in an urban location. A quite large part of the Introduction is spent on what fractions have been primary particles and originating from NPF in earlier studies in urban environments. Furthermore, since you claim that you can distinguish NPF events from primary emissions in this study, I think it should be more clear exactly how you define these two processes. In the Introduction, in lines 18-20 on page 2, you discuss production mechanisms of ultrafine particles from traffic: "condensation of semi-volatile phases vapor species that creates new UFP during dilution and cooling of engine exhaust emissions near the source". After that you write "most studies consider them as primary", "or quasi-primary particles". How do you define these particles, that form by nucleation in the tailpipe or a second after exiting? I guess you define them as primary, but I think that should be clear.**

We agree. It was not enough clear. We also define such particles as primary ones. The quoted paragraph has been rewritten as follows:

*In urban areas, traffic emissions are a major source of UFP (Kumar et al., 2014; Ma and Birmili, 2015; Pey et al., 2008; Pey et al., 2009; Dall'Osto et al., 2012; Salma et al., 2014; Paasonen et al., 2016). These emissions include primary UFP exhaust emissions (Shi and Harrison, 1999; Shi et al., 2000; Charron and Harrison, 2003; Uhrner et al., 2012); cooling of engine exhaust emissions and condensation of semi-volatile phases vapor species that creates new UFP during dilution (Charron and Harrison, 2003; Kittelson et al., 2006; Robinson et al., 2007; Rönkkö et al., 2017). These are also considered primary particles, since they are formed near the source.*

2.) **Comment 1) leads to the question how you know that the regional NPF events are not "primary aerosol particles" formed by nucleation in the tailpipe or soon after exiting. Such emissions of 6-11 nm particles (Kittelson et al., 2006) dominate the number emissions. Furthermore, such emissions from thousands of cars in the Madrid area would likely look like a regional NPF event, since the emitted particles will grow by condensation in the atmosphere. One argument against this hypothesis, that the particles are primary, is the fact that your formation rates and number concentrations of 9-25 nm particles peak at noon when BC levels are at minimum. On the other hand, condensational growth is strongest at noon or in the afternoon since photochemical production of condensable vapors is dependent on solar radiation. Therefore, even though primary emissions are likely highest in the morning and evening rush hours, the likeliness that the emitted nano-particles grow into the 9-25 nm interval (which you refer to as ultrafine particles and for which you present the diurnal cycle in Fig. 5a) may be highest at noon or in the afternoon. I think you need to strengthen your arguments and definitions here when you refer to the events as "regional NPF events". Also, could you add diurnal cycles of number concentrations associated with the PSM data to Fig. 5a?**

We consider regional NPF events days in which the particle size distributions at all our stations have the same evolution. Being these stations 17 km apart and being all of them of different categories (urban, urban background and suburban), we assume that the particles are not formed in the tailpipe, i.e., they are not primary. Otherwise there would be significant differences when comparing the PSD at the suburban station, which is not highly influenced by traffic, as opposed to the urban station. Another argument is the fact that particle concentrations measured with the PSM are higher in the suburban station, compared with the urban station (as shown in Fig S7). Additionally, PSM data show an increment in the concentration of all size ranges around the same time in which we begin to see growth in the SMPS size distributions, meaning that these particles started growing from 1.2 nm, therefore they are not 6-11 nm emissions as suggested. Dall'Osto et al. (2013) characterized at regional scale (40 km) the simultaneous occurrence of these photochemical nucleation events covering vast zones with very high nucleation. Because it was not clearly augmented, the following text was modified to clarify this question:

*In the selected episodes, intensive daytime nucleation and subsequent condensational growth processes took place simultaneously at urban and suburban stations, located 17 km apart, and accordingly we classify these as regional NPF episodes. Being all stations differently influenced by traffic (influenced, slightly and not influenced by traffic), we can affirm that these episodes are regional events and not representative of primary emissions. Otherwise, we would observe significant differences at our stations. Additional arguments are the fact that number concentrations of sub-25 nm particles peak at noon, when BC levels are at their minimum, as well as higher concentration of particles measured by PSM at the suburban station, compared to the urban station, implying that the particles are not originated from traffic sources.*

Fig. 5 (now Fig. 3) has been modified to include PSM data, as suggested by both referees. We agree that this figure was necessary. Now we see the 3 peaks, 2 from traffic and the midday one.

**3.) Following up on Comment 2), in Sect. 3.2.1 you equate the occurrence of "10 nm particles" with NPF. Could these particles not just as well be primary?**

See reply to Comment 2).

**4.) Page 5, lines 22-23: Why did you choose the number concentration in the 9-25 nm interval in Eq. 4 for your definition of the formation rate? Why do you not use your PSM data for the formation rates?**

We chose this size interval (9-25 nm) due to the detection limit of the SMPS, which is the instrument we used to calculate growth rates, sinks and formation rates. PSM data was very noisy and we preferred not to use it for this purpose. However, following both Referees' comments, the calculations have been made again using PSM data. Figure 6 and previous results using SMPS have been removed, and GR calculated with PSM data and $J_1$ results and discussion have been provided:

*Growth rates ($GR_{PSM}$) and total formation rates of 1.2-4.0 nm particles ($J_1$) were calculated from PSM data at CSIC and ISCIII stations. $GR_{PSM}$ were calculated from 11 to 18 July 2016, averaging 4.3 nm $h^{-1}$ at the urban station and 3.7 nm $h^{-1}$ at the suburban station. $J_1$ were calculated only for the days in which NPF is identified. The results for these days are included in Table 1. Average $J_1$*

*values are higher at the urban station (8.9 cm$^{-3}$ s$^{-1}$) compared to the suburban station (5.3 cm$^{-3}$ s$^{-1}$). Concentrations of 1.2-4.0 nm particles are lower at the urban station (Figure S6), which could lead to lower formation rates. However, the coagulation sink is greater at the urban station, as discussed before, which contributes to the second factor in Eq. (4). It has to be noted that only 3 days of PSM data were available for NPF events at the urban station. A longer dataset could lead to different results.*

*The average values of the formation rates agree with those reported at similar stations around the world. For instance, Woo et al. (2001) reported J$_3$ ranging 10-15 cm$^{-3}$ s$^{-1}$ in Atlanta, US. Wehner and Wiedensohler (2003) reported average J$_3$ of 13 cm$^{-3}$ s$^{-1}$ in Leipzig, Germany. Hussein et al. (2008) reported nucleation rates (D$_p$<25 nm) ranging 2.1-3.0 cm$^{-3}$ s$^{-1}$ in summer in Helsinki.*

**5.) I do not follow the conclusion written in the abstract on lines 37-39: "The vertical soundings demonstrated that ultrafine particles (UFP) are transported from surface levels to higher levels, thus newly formed particles ascend from surface to the top of the mixing layer", or that the fluxes are "bottom-up" as written on page 9, line 28. As far as I can tell, Fig. 10 (which is a very nice figure by the way) only shows that the particles are produced inside the mixed layer (not the residual layer). The mixed layer of course grows during the day, but how can you tell whether the particles are being produced close to the surface or at the top of the mixed layer (or both)?**

We agree with this statement. It was not clear enough. We can only say that the particles are produced inside the mixed layer, which grows during the day. We cannot tell if these particles are produced in a specific altitude inside this layer. This has been clarified in the text.

*The vertical soundings demonstrated that ultrafine particles (UFP) are formed exclusively inside the mixed layer. As convection becomes more effective and the mixed layer grows, UFP particles are detected at higher levels.*

**6.) Regarding shrinkage in lines 1-2 on page 7: Could evaporation be a reason for the shrinkage as well?**

Yes, in fact this is the case. Upon closer inspection of the wind speed and wind direction time series, we determined that wind speed increased during the shrinking phase, but the wind direction did not change substantially, therefore there is no change of air masses and the leading process is dilution, which favors evaporation. The text has been modified as follows:

*The start of the shrinking phase coincides with a marked increase in wind speed, therefore it is associated with dilution, which favors the evaporation of semi-volatile vapors, resulting in a decline in particle diameter and concentrations, as observed in most cases (see Figure 1).*

**7.) Page 7, lines 30-31: "For these stations the observed median growth rates were 7-8 nm h-1". These values are very close to the average growth rate of 7.3 nm h-1 reported from Bakersfield, a polluted location in California (Ahlm et al., 2012). Please also add this reference to the studies of NPF events in urban environments in lines 25-30 on page 2.**

This reference has been added.

**8.) Page 9, line 8. It seems this is the first time you discuss Fig. 3 so perhaps you should change the order of the figures.**

The order of the figures has been modified and this has been corrected.

**9.) I suppose the black dots in Figs. 7-9 represent the fitted log-normal modes, but please add that information to the figure captions.**

Yes. This has been added in the corresponding figure captions (now Figs. 4-6).

*Figure 7: Particle size distribution with fitted log-normal modes (black dots) measured during the balloons soundings at Majadahonda on 12 July 2016.*

**10.) You don't draw any conclusions from Fig. 9 so perhaps that figure is not necessary.**

We think that Fig. 9 (now Fig. 6) gives an additional confirmation of the fact that the particle formation takes place exclusively inside the mixed layer. The figure has been modified following Referee 2 suggestions to include an estimation of the mixed layer height, which adds significance to the figure. We also believe it makes it easier to understand Fig. 10 and we would prefer to keep it.

**References**

Ahlm, L., Liu, S., Day, D. A., Russell, L. M., Weber, R., Gentner, D. R., Goldstein, A. H., DiGangi, J. P., Henry, S. B., Keutsch, F. N., VandenBoer, T. C., Markovic, M. Z., Murphy, J. G., Ren, X., and Scott, S.: Formation and growth of ultrafine particles from secondary sources in Bakersfield, California, J. Geophys. Res. Atmos., 117, D00V08, doi:10.1029/2011JD017144, 2012.

Dall'Osto, M., Querol, X., Alastuey, A., O'Dowd, C., Harrison, R.M., Wenger, J., Gómez-Moreno, F.J.: On the spatial distribution and evolution of ultrafine particles in Barcelona. Atmos. Chem. Phys. 13, 741-759, 2013.

Kittelson, D.B., Watts, W.F., Johnson, J.P.: On-road and laboratory evaluation of combustion aerosols – Part1: Summary of diesel engine results. J. Aerosol Sci. 37, 913- 930, 2006.

**REPLY TO #2 REFEREE'S QUERIES AND DESCRIPTION OF CHANGES DONE FOLLOWING HER/HIS SUGGESTIONS**

Many thanks for your valuable comments and suggestions than helped us to improve the quality of the manuscript. As you will see we took into account all your comments when reviewing the new version.

**Major comments**

- **Despite the interesting dataset this paper fails in conveying a clear message due to its structure and the way the results are presented. In particular the authors mix the main results (horizontal and vertical characterization of NPF in Madrid) with episodes (e.g. the UFP peaks at night) and phenomena (e.g. shrinkage of particle size) that are not strictly related with them. To address this issue I suggest to the authors to focus more on the important results and emphasize them in a clearer way as well as to restructure the results section in order to make a clear separation between these results and all the minor observations.**

Thanks a lot for your comments. This suggestion has been taken into account and section 3 has been reorganized as follows:

3.1 Meteorological context
3.2 Comparison of NPF events at urban and suburban stations
       3.2.1 Episode characteristics
       3.2.2 Comparison of GR, $J_1$, CS and $CoagS_9$
3.3 Vertical distribution of NPF events
       3.3.1 UFP concentrations
       3.3.2 Particle size distribution
3.4 Other observations
       3.4.1 Prevalence of particles and shrinking
       3.4.2 Nocturnal UFP peaks

- **The authors state several times that NPF is dominating the total particle concentration in Madrid, however it is not clear how they separate between newly formed particles and ultrafine particles directly emitted from cars. I assume that the formation rate calculation is biased by the fact that primary UFP were not taken into consideration and this would explain why formation rates at the urban stations are higher than those measured in the suburban whereas the growth rates are smaller. The authors need to quantitatively estimate the source of UFP and revise the formation rates excluding primary UFP from their results. Doing this would also allow to compare directly the Madrid case with the other locations reported in the introduction. Probably the simplest way to discriminate between primary and secondary UFP would be to use the data of the particle concentrations below 9 nm, that should be available at all the measurement sites.**

Thanks again for this comment. If both referees commented on this, it is because it was not clear enough. We have followed both referees' suggestions to address this issue and we have calculated formation rates using PSM data (see replies to following comments). We have also clarified in the text that we consider NPF events days in which the particle size distributions at all our stations have the same evolution. Being these stations 17 km apart and being all of them of different categories (urban, urban background and suburban), we assume that the particles are not formed in the tailpipe,

i.e., they are not primary. Otherwise there would be significant differences when comparing the PSD at the suburban station, which is not highly influenced by traffic, as opposed to the urban station. Another argument is the fact that particle concentrations measured with the PSM are higher in the suburban station, compared with the urban station (as shown in Fig S7). Additionally, PSM data show an increment in the concentration of all size ranges around the same time in which we begin to see growth in the SMPS size distributions, meaning that these particles started growing from 1.2 nm, therefore they are not 6-11 nm emissions as suggested. Dall'Osto et al. (2013) characterized at regional scale (40 km) the simultaneous occurrence of these photochemical nucleation events covering vast zones with very high nucleation. The following text was modified to clarify this:

*In the selected episodes, intensive daytime nucleation and subsequent condensational growth processes took place simultaneously at urban and suburban stations, located 17 km apart, and accordingly we classify these as regional NPF episodes. Being all stations differently influenced by traffic (influenced, slightly and not influenced by traffic), we can affirm that these episodes are regional events and not representative of primary emissions. Otherwise, we would observe significant differences at our stations. Additional arguments are the fact that number concentrations of sub-25 nm particles peak at noon, when BC levels are at their minimum, as well as higher concentration of particles measured by PSM at the suburban station, compared to the urban station, implying that the particles are not originated from traffic sources.*

- **In section 2.2 no details are provided about the sampling conditions. The authors should give a short description of the inlets and explain for example if losses were measured and/or calculated, if any intercomparison between the different particle counters was performed, etc... Moreover, a big part of the work is based on the measurements performed with the Hy-SMPS but no proper characterization is provided (Figure S2 is not really useful to evaluate the performances of this instrument) and the only cited paper is written in Korean (I'm also not sure whether this is a peer reviewed journal or not). For these reasons a more complete characterization of this instrument is required, for example it could be useful to compare the Hy-SMPS with a reference SMPS while looking at separate size bins and not only at the total concentration.**

TSI instruments were corrected for diffusion losses and multiple charge losses using the instruments' own software. The measuring conditions were the same at all sites. PSM data was post-processed using tailored software provided by Airmodus.

The journal *Particle and Aerosol Research* is published by the Korean Association for Particle and Aerosol Research (KAPAR). It is a peer reviewed journal. A comparison between Hy-SMPS and a TSI SMPS is provided in the supplementary material. Figure S2 has been modified and the following text has been modified in page 2 line 38:

*The instrument was intercompared with a TSI-SMPS (Standard DMA with 3776 CPC) for 50-nm monodisperse NaCl particles and polydisperse aerosol (Fig. S2).*

- **In section 2.3 it is explained how formation and growth rates are calculated, however no explanation is provided about the decision of using the 9-25 nm range, despite the fact that measurements of particle concentrations down to about 1nm were performed. I would argue that this is not the best choice, in particular for the formation rates that could be highly biased by primary emissions as previously explained. Moreover, using a**

**smaller reference diameter for the formation rate would permit to better compare with measurements performed in other locations and/or in chamber studies. For these reasons I would suggest to calculate formation rates for a more meaningful size (below 5 nm) and eventually to calculate growth rates at two different sizes (for example keep the 9-25 nm range for comparability with the vertical sounding and add a lower size GR depending on the availability of the existing data). Finally, uncertainties for all the growth and formation rates should be estimated in order to make a proper comparison between different locations and days.**

We chose this size interval (9-25 nm) due to the detection limit of the SMPS, which is the instrument we used to calculate growth rates, sinks and formation rates. PSM data was very noisy and we preferred not to use it for this purpose. However, following both Referees' comments, the calculations have been made again using PSM data. Figure 6 and previous results using SMPS have been removed. Calculated growth rates and formation rates with PSM have been added to the discussion:

*Growth rates ($GR_{PSM}$) and total formation rates of 1.2-4.0 nm particles ($J_1$) were calculated from PSM data at CSIC and ISCIII stations. $GR_{PSM}$ were calculated from 11 to 18 July 2016, averaging 4.3 nm $h^{-1}$ at the urban station and 3.7 nm $h^{-1}$ at the suburban station. $J_1$ were calculated only for the days in which NPF is identified. The results for these days are included in Table 1. Average $J_1$ values are higher at the urban station (8.9 cm$^{-3}$ s$^{-1}$) compared to the suburban station (5.3 cm$^{-3}$ s$^{-1}$). Concentrations of 1.2-4.0 nm particles are lower at the urban station (Figure S6), which could lead to lower formation rates. However, the coagulation sink is greater at the urban station, as discussed before, which contributes to the second factor in Eq. (4). It has to be noted that only 3 days of PSM data were available for NPF events at the urban station. A longer dataset could lead to different results.*

*The average values of the formation rates agree with those reported at similar stations around the world. For instance, Woo et al. (2001) reported $J_3$ ranging 10-15 cm$^{-3}$ s$^{-1}$ in Atlanta, US. Wehner and Wiedensohler (2003) reported average $J_3$ of 13 cm$^{-3}$ s$^{-1}$ in Leipzig, Germany. Hussein et al. (2008) reported nucleation rates ($D_p$<25 nm) ranging 2.1-3.0 cm$^{-3}$ s$^{-1}$ in summer in Helsinki.*

- **Section 3.2.2 reports a comparison of growth rates and formation rates at the different sites but without including the vertical profiles (in this case growth rates are provided in section 3.3). I think that the paper would benefit by having all the growth rates presented together, this would improve the readability and the clarity of the paper.**

All growth rates, including those of the vertical soundings, were already provided in Table 1, in section 3.2. However, following your suggestion, we have added this information in the discussion of the surface stations to improve the clarity.

*Figure 2 shows the growth rates presented in Table 1 according to urban and suburban surface stations. GR regarding the vertical measurements are provided in the following section due to differing sampling periods.*

- **Section 3.2.3 reports PTR measurements of 3 ions that show some correlation with the particle growth, however this section is not adding any valuable information to the overall picture of the paper. For this reason I would suggest to either remove it or expand it with more detailed analysis. For example it would be worth investigating the possible**

**precursors for HOMs formation. It may be possible to say something about the origin of the condensable vapours by looking at the concentration and diurnal profiles of biogenic vs. anthropogenic VOCs.**

We conceived section 3.2.3 as a brief presentation of interesting preliminary results rather than a central part of the paper. We think that expanding this section is not reasonable, considering both the amount of work needed, and the fact that it is not an essential part of this paper. However, we believe it is worth it to move these results to the supplementary information.

- **Section 3.2.4 should be written in a more consistent way, in particular the authors first speak of sub-25 nm particles but then they only show bivariate plots of sub-4nm particle concentration, what is the reason for this? Moreover to better support the airport hypothesis it would be useful to show a time series of UFP for the period of interest together with wind speed and wind direction (extrapolating these data from the supplementary is difficult due to the long time period reported there). I'm asking this because the bivariate plots only show that UFP particle concentration is higher when the wind is coming from NE but to support the authors hypothesis it would be important to check if there are periods with low UFP concentration under the same wind conditions. If this is the case then I wound find the hypothesis less convincing due to the fact that the airport should be a more or less stationary source of UFP.**

We considered that it was interesting to see if these particles were growing from smaller diameters to better determine their origin, hence the choice to use sub-4nm particle concentration instead of sub-25nm. This has been clarified in the text. We followed your suggestion to better support our hypothesis, and we have included an additional figure in the SI showing PSM data together with wind direction and wind speed, highlighting periods with NE direction and high wind speed, as well as periods with low particle concentration. A version of this figure is presented in Figure D1. The episodes described in the manuscript (12-14 July) coincide with NE directions and high wind speed. During the sampling period, there are not other periods in which these two conditions apply simultaneously. Therefore, according to the available data, there are no periods with low UFP under the same wind conditions. The following text has been added to the discussion in the corresponding section:

*To better support this hypothesis, Fig. S7 shows PSM data together with wind direction and wind speed, showing that the episodes coincide with strong NE winds, and that there are not episodes with low UFP concentrations with these same conditions.*

[Figure]

Figure D1: Concentration of particles >2 nm measured with PSM at CSIC station, wind direction and wind speed from 10 to 20 July 2016. $N_2$ lower than 25th percentile has been highlighted, as well as NE directions and wind speeds higher than 4 m/s.

- **In section 3.3 the authors often speak of "bottom-up flux" for the UFP however, what vertical profiles show is only that the UFP concentration is homogeneous inside the mixing layer. For this reason one should avoid using this terminology and the authors should correct all the corresponding parts in the paper. Moreover, I found that the interpretation of the vertical profiles graphs is complicated by the absence of a direct measurement of the mixing layer height. Referring to the "twin paper" [1], this information should be available for all the soundings (for example the potential temperature or total particle concentration can be used) and it would be a really nice addition to the graphs.**

We agree with this statement. We can only say that the particles are produced inside the mixed layer, which grows during the day. We cannot tell if these particles are produced in a specific altitude inside this layer. This has been clarified in the text.

A rough estimation of the mixing layer height has been added to the plots as suggested. The following text has been added in sections 2 and 3:

*Particle concentration in the range 3-1000 nm was measured with a miniaturized butanol-based CPC (Hy-CPC). The time resolution was 1 s, and sample flow was 0.125 L/min (Lee et al., 2014).*

*A rough estimation of the mixed layer height was determined using Hy-CPC measurements. The top of the mixed layer was considered at an altitude in which particle concentration decreases an order of magnitude quasi-instantaneously and remains constant above. All UFP profiles are included in Querol et al. (2018).*

*The interphase between the mixed layer and the residual layer, i.e. the mixed layer height, has been derived using the UFP vertical profiles (see Querol et al., 2018).*

- **The conclusions are written as a summary of the paper but here the author should focus more on the significance of their findings compared with existing observations. This section should be rewritten in order to convey a clearer message, so I suggest to the authors to delete all the unnecessary parts focusing more on the important results of the paper.**

Following your directions, we have re-written the conclusions as follows:

*We investigated the phenomenology of regional and secondary New Particle Formation (NPF) episodes in central Spain. To this end we set up 3 supersites (an urban, a urban background and a sub-urban background) 17 km away in and around Madrid. We were able to characterize 6 NPF events, and in all cases the evolution of the particle size distribution (PSD) was very similar at all stations: around sunrise nucleation mode particles appear and start growing and in the afternoon a decline in particle sizes, i.e. shrinkage, is observed. The regional origin of the NPF is supported by the simultaneous variation in PSD in the nucleation mode and particle number concentrations, growth and shrinkage rates. Furthermore, time trends of condensation and coagulation sinks (CS and CoagS) were similar at all stations, having minimum values shortly before sunrise and increasing after dawn towards the maximum value after midday in the early afternoon. In spite of the 17 km scale simultaneous processes affecting particle number concentrations, the following relevant differences between urban and suburban stations were observed: i) the urban stations presented larger formation rates and smaller growth rates as compared to the suburban stations; ii)in general, the sinks were higher at the urban stations.*

*Regarding the vertical soundings of the NPF events, we observed that in the early morning the vertical distribution of newly formed particles is differentiated in two layers. The lower layer (mixed layer, ML) in which convection is effective, is well-mixed and has a homogeneous PSD. This ML heightens throughout the day, as insolation is more pronounced, extending beyond the sounding limits around midday. NPF occurs throughout this ML, and growth rates and concentrations are homogeneous. The upper layer is a stable residual one (RL) in which particles formed or transported the previous days prevail. In the RL growth is inhibited or even completely restrained, compared with the same particles in the ML. Overall, the soundings demonstrate that particles are formed inside the ML, but they can prevail and be displaced and stored at upper levels and continue to evolve on following days.*

*Additionally, a few nocturnal bursts of nucleation mode particles were observed in the urban stations, which could preliminarily be related with aircraft emissions transported from the airport of Madrid.*

*In this campaign we could not measure in the earliest stages of NPF due to safety requirements of the balloon flights early in the morning. We think it is important for future work to carry out soundings during the nucleation phase of the episodes. However, miniaturized instruments able to measure smaller particles would be needed, which are not available at the present time. This would allow us to determine whether secondary NPF takes place throughout the ML or occurs at the surface and is transported upwards by convection afterwards. If the former were true, then locations with high ML could produce more secondary particles than we have considered, and they could affect a larger population, or influence climate to a greater extent.*

*We cannot determine whether the NPF episodes were triggered by the pollution generated in the city that extended to the region, or the events are caused by a broader phenomenon. In either way, it can be concluded that in summer the particle number concentrations are dominated by NPF in a wide area. The impact of traffic emissions on concentrations of UFP is much smaller than those of NPF, even near the city center where the pollution load is at the highest. This result is in line with other studies performed in cities from high insolation regions (e.g. Kulmala et al., 2016). Given the extent of the episodes, the health effects of NPF can affect a vast number of people, considering that the Madrid metropolitan area with more than 6 million inhabitants is the most populated area in Spain, and one of the most populated in Europe (UN, 2008). For this reason, we believe that the study of health effects related to newly-formed particle inhalation is crucial.*

**Minor comments**

- **Page2 line 3: "The NPF events extend over the full vertical extension of the mixed layer reaching as high as 3000 m." But the maximum height of the sounding is 2000 m, so this should be corrected.**

Although the maximum height of the sounding is 2000 m, we state that the events take place over the full extension of the mixing layer, which other authors (e.g. Plaza et al., 1997) have found to reach as high as 3000 m. This sentence has been rewritten to clarify this.

*The NPF events extend over the full vertical extension of the mixed layer, which can reach as high as 3000 m in the area, according to previous studies.*

- **Page 2 line 4: "This can have consequences in the radiative balance of the atmosphere and affect the climate", the climatological effect of NPF in a polluted environment as Madrid is questionable and not supported by any evidence in this paper, for this reason remove or rephrase this sentence.**

We agree with this comment. Since we do not provide any evidence for this we have removed this sentence.

- **Page 2 line 5: As previously stated, a proper estimation of NPF over primary UFP should be given here.**

See reply to the second major comment.

- **Page 2 line 25: There is no need to cite 25 papers, this can be reduced.**

We have reduced the number of citations.

- **Page 3 line 13: The sentence seems to contradict the cited paper of Querol et al.[1] where it is said that "Relatively low concentrations of ultrafine particles (UFPs) were found during the study, and nucleation episodes were only detected in the boundary layer."**

On page 3 lines 13-15 it is said that "intensive NPF episodes take place inside the planetary boundary layer (PBL) in Barcelona, occurring around midday at surface levels when insolation and dilution of pollution are at their maxima". This agrees with the statement quoted from [1].

- **Page 4 line 13: It would be useful to add a scale to the map in figure S1.**

Figure S1 has been modified accordingly.

- **Page 4 line 19: It is not clear if the PSM was operated in scanning mode or fixed mode, it would be good to specify this.**

The PSM was operated in scanning mode. This has been specified in the text.

*[…] and a Particle Size Magnifier (PSM) (AirModus) in scanning mode for the size range 1.2-2.5 nm.*

- **Page 6 line 14: It is said that NPF was identified on 12 days but table 1 only reports 7 days so this should be made consistent. Moreover, Figure S8 shows that there were 2 nucleation events at CIEMAT on the 13th and 14th of July that are not mentioned in the table. In addition, it would be useful to add also formation rates to the table together with the GR values.**

This is a mistake, we wanted to say that we selected 12 episodes amongst the 18 the identified episodes, considering all stations, occurring on 6 days (7 days identified). We have rewritten this sentence to make it clear:

*18 NPF episodes have been identified on a total of 7 days throughout the campaign. In Table 1 a summary of these events is presented. Out of these, a total of 14 events on 6 days had simultaneous data available for at least one of the urban stations (CSIC, CIEMAT) and the suburban station (ISCIII).*

We have added formation rates $J_1$ to the table as suggested.

- **Page 6 line 27: It's almost impossible to see the early morning UFP concentration just by looking at the full time series. Thus a dedicated plot should be made either by plotting sub-10nm particles on top of Figure 1 or by plotting a diurnal profile for this size range.**

Figure 5 (now Fig. 3) has been modified to include a diurnal profile of total particle concentration using PSM data, as suggested in other comments.

- **Page 7 line 1: Particle size shrinking is an interesting phenomenon but it doesn't fit nicely in this part of the text. For this reason describe it in a separate (small) section. It would also be nice to plot the wind speed specifically for the shrinking phase because from the overall plot it is difficult to see a trend. In some cases also a rapid change in number concentrations is observed pointing to a change in air mass. In these cases it does not make sense to speak about a shrinking of aerosols because the aerosol population changes. The authors should demonstrate clearly, that the same population of aerosols is shrinking. Otherwise, it is not shrinking and they should delete this.**

We agree with this comment. Following your suggestions, we have moved the shrinking discussion to a short section.

Upon closer inspection of daily plots of wind speed and wind direction (Fig. S7), we determined that wind speed increased during the shrinking phase, but the wind direction did not change substantially, therefore there is no change of air masses and the leading process is dilution, which favors evaporation. The text has been modified as follows:

*The start of the shrinking phase coincides with a marked increase in wind speed, therefore it is associated with dilution, which favors the evaporation of semi-volatile vapors, resulting in a decline in particle diameter and concentrations, as observed in most cases (see Figure S7).*

This new figure has been added to the supplementary information (Fig. S7).

[Figure]

Figure S7: Daily plots of wind speed and wind direction for the days in which shrinkage is observed.

- **Page 7 line 14: Here a reference to Figure 5 would be useful. Moreover, I think that associating UFP with particles in the range 9-25 nm is misleading and it would be better to plot the diurnal profile for all particles below 25 nm and for the total particle concentration.**

Following the first comment we have restructured the text and this is not an issue any more.
As mentioned in the previous comment, daily averages of total particle concentration using PSM data have been added to Fig. 5.

- **Page 8 line 5: "above" should be replaced by "below".**

This has been replaced by a reference to the section in which the results are discussed.

- **Page 8 line 11: "The fact that J9 is higher at the urban stations is probably linked to higher traffic emissions [...] in the city, and not related with higher nucleation rates, since PSM measurements indicate lower concentrations of 1.2-4 nm particles". Here the authors confirm my hypothesis that primary particles affect formation rates, underlining the necessity to take this process into account in their calculation.**

As mentioned above, formation rates have been calculated again using PSM data to evaluate this fact and the corresponding results have been added to the discussion.

- **Page 8 line 14: This is not figure S4 but figure S7.**

The order of the figures has been revised and this has been corrected.

- **Page 8 line 14: "The calculated formation rates agree with those reported in other studies, ranging 0.01-10 cm-3s-1 during regional events around the world." I don't see any reason for reporting an agreement within 4 orders of magnitudes. The formation rates measured during this campaign should be compared in a more targeted way with other locations around the world.**

Once formation rates have been calculated with PSM data this has been revised as follows:

*With average values of 8.9 and 5.3 $cm^{-3}$ $s^{-1}$ at the urban and suburban station respectively, the calculated formation rates agree with those reported at similar stations around the world. For instance, Woo et al. (2001) reported $J_3$ ranging 10-15 $cm^{-3}$ $s^{-1}$ in Atlanta, US. Wehner and Wiedensohler (2003) reported average $J_3$ of 13 $cm^{-3}$ $s^{-1}$ in Leipzig, Germany. Hussein et al. (2008) reported nucleation rates ($D_p$<25 nm) ranging 2.1-3.0 $cm^{-3}$ $s^{-1}$ in summer in Helsinki.*

- **Page 8 line 30: "Thus, the particles growth appears to be driven by the uptake of secondary organic compounds." This is a reasonable assumption but cannot be proven by the PTR measurements presented in this work. Try to support this assumption with additional information, for example one can try to check if the growth rates can be explained by sulfuric acid alone (assuming a reasonable range of values for sulfuric acid concentration) or not.[2, 3]**

We don't have the data needed to support this assumption. We have stated in the text that $SO_2$ levels were below the detection limit of the standard air quality UV spectrometry instruments during all the period. This has been added to the text, which has been moved to the SI as stated in a previous comment.

*We cannot prove this assumption using the PTR-ToF-MS measurements. We cannot check if the growth rates can be explained by sulfuric acid alone, since $SO_2$ levels were below the detection limit of the standard air quality UV spectrometry instruments during all the period.*

- **Page 10 line 10: "the mode slightly decreases its size when the sounding ascends above the mixed layer limit", as already written above the mixing layer height should be plotted together with the particle size distribution to better visualize these changes.**

An estimation of the mixed layer height has been provided in the corresponding figures.

- **Page 10 line 12: It would be good if the authors could specify how they calculated the growth rate in the residual layer. The impressions from the graphs is that there are really few points inside the residual layer and it is not clear whether there is any growth at all inside this layer.**

The growth rate here was calculated in the same way that in the mixed layer, selecting only the points inside the residual layer. Even though it might be not clear in the graphs, there are enough points to carry out the calculations when zooming in into the short time period of interest. In the presented graphs the dots are overprinted and it might look like there are not enough points to carry out these calculations.

- **Page 10 line 15: I'm not really convinced by the presence of a 10 nm mode in the first sounding, Maybe there is an over-fitting issue with the mode fitting algorithm. For this reason I think calculation of the growth rate is questionable and should be avoided.**

We believe that this is not an over-fitting. The same mode is observed simultaneously at the nearby (<3 km) ISCIII station at surface level using TSI instruments. We added this to the text to justify the calculation.

*Moreover, during the morning we observed particles growing inside the mixing layer from 10 nm at 7:00 UTC, to 30 nm at midday, with a growth rate of 3.5 nm h$^{-1}$. This mode is observed simultaneously at ISCIII and therefore we consider it for calculation. The growth rate obtained is 3.5 nm h$^{-1}$.*

- **Page 10 line 23: "The accumulation mode grows from 156 nm at 07:00 UTC to 200 nm at 10:00 UTC", also in this case I think the accumulation mode is over-fitted, the authors should either revise their fitting algorithm or prove that I'm wrong by reporting in the SI a single SMPS scan plot with the fitted modes on top of it.**

We agree, we removed all the results regarding the accumulation mode because we cannot prove that the fitting is correct for the accumulation mode.

- **Page 10 line 26: "Another mode starting roughly at 40 nm at 09:00 UTC" I guess this should be 07:00 UTC.**

Yes, this has been corrected.

- **Page 10 line 30: I don't see any nucleation mode earlier than 09:30-10.00 UTC. Correct this sentence and eventually revise the calculated growth rate.**

The fitting algorithm considers the appearance of the nucleation mode by 8:00 UTC, but we agree that we can't consider it until at least 9:00 UTC. Comparing with other stations, we considered that the mode appears at 9:00 UTC and calculated all growth rates from that time. The sentence has been rewritten to clarify this.

*A nucleation mode grows from the detection limit of the instrument, around 10 nm at 08:30 UTC to 40 nm at 15:00 UTC. Comparing with other stations, we considered this mode only after 9:00 UTC, and calculated the growth rates from that time. We consider this a regional NPF event, since the start of the particle growth is registered simultaneously at all the stations. The growth rates at the sounding location, ISCIII and CSIC are 5.3 nm h$^{-1}$, 4.6 nm h$^{-1}$ and 2.0 nm h$^{-1}$, respectively.*

- **Page 10 line 38: "As the insolation increased, so did the altitude of the mixing layer, until it reached the altitude at which the balloons were positioned." By looking at the plot it seems more likely that the balloon height decreased until reaching the mixing layer.**

The tethered balloons were positioned at a fixed altitude, meaning that the extension of the wire was not modified during these flights. However, wind conditions can vary the altitude of the instruments – for example increasing or decreasing horizontal wind speed – and this is what we see from 9:30 to 10:30 UTC in Fig. 9 (now Fig. 6). We have modified the text to explain this.

*In order to verify this result two constant altitude flights were made during the morning. The extension of the wire was not modified during these flights. However, changing wind conditions varied slightly the altitude of the instruments. The altitude was chosen so that the instruments remained initially outside the mixing layer, i.e. inside the residual layer.*

- **Page 10 line 40: As previously explained avoid speaking of particles flowing upward, measurements are just showing that UFP are homogeneous inside the mixed layer.**

This has been corrected accordingly as follows:

*As the mixing layer reached the balloons, total particle concentration sharply increased from $4x10^3$ to $2x10^4$ $cm^{-3}$, demonstrating that newly-formed particles remain inside the mixing layer.*

- **Page 11 line 5: I do not see a growth of 40 nm particles in the residual layer. The size of this mode is the same at the 9 and 11 UTC sounding.**

There is growth of this mode, revealed by the growth rate calculation. This might not be evident visually due to the use of a log-scale. However, the text has been clarified to make it clearer.

*Inside the residual layer particles had a slower growth rate (0.5 nm $h^{-1}$ compared to 8.45 nm $h^{-1}$ for the 40 nm mode – note that due to the use of a log-scale this might be unnoticeable visually), and no particles smaller than 20 nm were observed.*

- **Page 11 line 6: how do you know that you observed these particles already the previous day? Moreover, also here I think that the Aitken mode is over-fitted.**

There is a mistake in this sentence. We were referring to the accumulation mode. Following other comments, we removed all results regarding accumulation-mode particles and this sentence has been removed.

However, this comment applies to the discussion of Fig. 7. We know that they are the same particles by definition of the residual layer:
About a half hour before sunset the thermals cease to form, allowing turbulence decay in the formerly well-mixed layer. The resulting layer of air is called residual layer because its initial mean state variables and concentration variables are the same as those of the recently-decayed mixed layer. […] The residual layer often exists for a while in the mornings before being entrained into the new mixing layer. [4]
Since an approximation of the mixing layer height has been provided following the suggestions of previous comments, we can affirm that this layer is above the mixing layer, and therefore it is the residual layer, which contains the particles that were observed in the mixed layer the day before. This has been stated in the text when discussing Fig. 7, and a reference was added.

*The fact that sub-40 nm particles are not detected at the higher levels of the first flights suggests that convection is not very effective yet, and the sounding goes through different atmospheric layers, most likely the mixed layer and the residual layer. In the residual layer Aitken-mode particles formed on previous days prevail (Stull, 1988).*

- **Page 11 line 16 and following lines: I would avoid speaking of the accumulation mode. I'm really sceptical about the presence of this mode in the measurements presented here and, even if it is present, then it is above the detection limit for most of the time. Moreover, it is said that the accumulation mode grew faster than the other modes and "this phenomenon has been rarely reported in ambient air." I think the data do not support this conclusion. If I'm not mistaken the fitted accumulation mode shows a growth only for a couple of hours on a specific day and the data are quite scattered so it doesn't**

**seem like the growth is significantly higher compared with the Aitken mode. If the authors want to support this observation then they should try to look if anything similar is present in the SMPS ground measurements.**

As mentioned in a previous comment, we have removed all the results involving the accumulation mode, since we cannot prove that this mode is not over-fitted.

- **Page 12 line 13,14: As already explained the vertical profiles do not show a clear accumulation mode and this is particularly true for the residual layer, so I would remove this sentence.**

See previous comment.

- **Page 12 line 18: the authors don't need a miniaturized instrument with "greater resolution" but an instrument able to measure smaller particles.**

We agree with this comment and it has been corrected in the text.

*However, miniaturized instruments able to measure smaller particles would be needed, which are not available at the present time.*

- **Figure 1: I would greatly recommend to avoid using jet colormap (i.e. rainbow colormap) for surface plots. This colormap is not perceptually uniform and this can create several kinds of issues as widely documented elsewhere (e.g. https://www.ncbi.nlm.nih.gov/pubmed/22034369 and http://ieeexplore.ieee.org/document/4118486/?reload=true)**

This figure has been replaced by the former Figure S8, because the latter contains the same information in a clearer presentation, in addition to information regarding total particle concentration, as suggested in another comment. However, a rainbow colormap is also used in the new figure because of the limitations of the software used to produce the plot (Igor Pro, WaveMetrics).
Following this suggestion, we have changed the colormap in previous Figure 10 (now Fig. 7), which was also using jet colormap.

- **In Figure S8 the authors report the total size distribution for the three measurement sites. This is a useful supplementary information but the readability of the graph should be improved. In particular a logarithmic color scale should be used as well as a higher image resolution. I also suggest to extrapolate the total particle number concentration for the 3 sites in the same size bins for better comparability rather than using different sizes for each site. Finally I noticed that there are some mismatches in the merged size distributions measured at CIEMAT (e.g. 8/7/2016). This would indicate that one of the instruments was not working properly. Please comment on this. How would this affect the presented results?**

The figure has been modified, using a logarithmic color scale as suggested, and the total particle number concentration is now used for the 3 sites.
The fact that there is a mismatch in the size distributions at CIEMAT is because the two instruments were measuring in different size ranges. The instrument measuring the smallest particles had more losses. This was corrected prior to the calculations; however, we didn't include the corrections in

this figure. Now the figure has been corrected to take this into account and a description of the corrections made has been added in section 2 as follows.

*Important discrepancies were observed after merging both SMPS particle size distributions. In order to correct that, we studied the distribution of particles in the coinciding size range (14-31 nm). The daily nanoSMPS size distribution was divided by the daily average of this range. We compared the resulting merged particle size distribution with CPC measurements, to check that there was a good agreement in the total particle concentration.*

**References**

[1] X. Querol et al. "Phenomenology of summer ozone episodes over the Madrid Metropolitan Area, central Spain". In: Atmospheric Chemistry and Physics Discussions 2017 (2017), pp. 1-38. doi: 10.5194/acp-2017-1014. url: https://www.atmos-chem-phys-discuss.net/acp-2017-1014/ (cit. on pp. 3, 4).

[2] T. Nieminen et al. "Sub-10 nm particle growth by vapor condensation effects of vapor molecule size and particle thermal speed". In: Atmospheric Chemistry and Physics 10.20 (2010), pp. 773-9779. issn: 16807316. doi: 10.5194/acp-10-9773-2010 (cit. on p. 5).

[3] Jasmin Tröstl et al. "The role of low-volatility organic compounds in initial particle growth in the atmosphere". In: Nature 533.7604 (2016), pp. 527-531. issn: 0028-0836. doi: 10.1038/nature18271. url: http://www.nature.com/doifinder/10.1038/nature18271 (cit. on p. 5).

[4] Roland B. Stull. "An introduction to boundary layer meteorology". Kluwer Academic Publishers, Dordrecht, Boston and London, 1988.

---

## Referee Report (RR1)

I would like to thank the authors for taking my comments into account. I think the manuscript has been much improved. I only have a few more comments. (Page and line numbers refer to the modified version of the manuscript with tracked changes.)

The reason that I suggested adding PSM data to Fig. 3 is to better separate new particle formation from primary emissions. However, the PSM total (>2.5 nm) number concentration also includes primary emissions. Could you add the concentration within the interval 1.2-2.5 nm to Fig. 3?

Page 1: line 30: Change "bigger" to "larger".

Page 8, lines 34-35: Please rephrase. One suggestion is: "if we assume equally large vapor sources at all locations, lower growth rates may be expected at the urban stations where the vapor will be distributed over a larger number of particles."

Page 11, line 29: Change "less vapors" to "lower vapor concentrations".

---

## Referee Report (RR2)

2 July 2018

I want to thank the authors for their work, the quality of the paper was improved and many issues addressed. However I still have 2 main concerns:

- In section 3.2.2 it is stated that growth rates in the suburban station are "significantly higher" than those in the urban area. I'm not convinced about the presence of a significant difference, in particular because the authors didn't provide any estimate on the uncertainties that affect their calculation (as I requested in my previous review). Moreover, table 1 shows that the highest Gr in the suburban station were measured on the 16/06 and 17/06 but from figure 1 is it evident that during these days there is not a clear growth and the Gr calculation can easily be biased by the absence of a defined growing mode. The authors should comment on this and eventually revise their statements.

- The argumentations provided in section 3.4.2 to support the hypothesis that the airport is the source of the nocturnal UFP peak are not strong enough in my opinion. I thank the authors for preparing figure S8, this is really helpful and confirms my initial scepticism. The nocturnal UFP peak lasted for about one hour on all the 3 days, whereas the wind conditions (NE direction, strong wind speed) stay more or less constant over several hours. Moreover, Madrid airport seems to have flights during all night long so the period with high UFP concentration should last much longer. I cannot see how this UFP peaks can be linked with the airport given the information provided in this manuscript.

  The authors should provide more convincing argumentation in favour of this hypothesis or look for other possible causes. I would like to stress about the importance of being careful with these kind of overstatements: this manuscript was highlighted on the airmodus webpage (https://airmodus.com/nocturnal-sub-3-nm-particles-in-madrid-airmodus-newsletter-22018/) with a title saying that the airport is a source of ultrafine particles in Madrid no matter the fact that the paper was still in the review phase and no strong evidences for this causation links were provided.

In addition to these 2 major comments i also have a couple of less important considerations:

- Page 4 lines 16-20: I don't really understand what the authors mean here. They mentioned that important discrepancies were found but I don't understand how this was taken into account. The authors should rephrase this part or add a couple of sentences to explain it better.

- Page 4 line 29-30: In my previous review I asked to provide more details about the sampling conditions, I appreciate that the authors added this sentence but I was expecting something more about the length/size of the inlets, the flow rates and the losses inside those lines. The authors should add a short description of their setup.

- Page 8 line 9: here it is said that "only 3 days of PSM data were available" however this is misleading because PSM data are available for a much longer period. What is missing is a longer overlapping between PSM and SMPS data so this sentence should be made more clear.

---

## Author Response (AR2)

**REFEREE #1**

I would like to thank the authors for taking my comments into account. I think the manuscript has been much improved. I only have a few more comments. (Page and line numbers refer to the modified version of the manuscript with tracked changes).

The reason that I suggested adding PSM data to Fig. 3 is to better separate new particle formation from primary emissions. However, the PSM total (>2.5 nm) number concentration also includes primary emissions. Could you add the concentration within the interval 1.2-2.5 nm to Fig. 3?

*REPLY: Thank you very much for your comments and new suggestions. We added this fraction to Fig.3. As you can see the concentration in this interval is also larger at the suburban station throughout the day.*

Page 1: line 30: Change "bigger" to "larger".

*REPLY: Done, thank you.*

Page 8, lines 34-35: Please rephrase. One suggestion is: "if we assume equally large vapor sources at all locations, lower growth rates may be expected at the urban stations where the vapor will be distributed over a larger number of particles."

*REPLY: Thank you for your suggestion. However, we finally deleted this sentence after reviewing this subsection as suggested by referee #2. With the reviewed results we cannot affirm that there is a difference in growth rates.*

Page 11, line 29: Change "less vapors" to "lower vapor concentrations".

*REPLY: Done, thank you.*

**REFEREE #2**

I want to thank the authors for their work, the quality of the paper was improved and many issues addressed. However I still have 2 main concerns:

*REPLY: Thank you very much for your comments and new suggestions.*

- In section 3.2.2
  - it is stated that growth rates in the suburban station are "significantly higher" than those in the urban area. I'm not convinced about the presence of a significant difference, in particular because the authors didn't provide any estimate on the uncertainties that affect their calculation (as I requested in my previous review).

*REPLY: The method used for the calculation of the growth rates (Dal Maso et al., 2005) always has a subjective factor, and for this reason it is made in a group of at least three experts in order to avoid biases. For this reason, the uncertainty can be important but are difficult to measure. Nonetheless, we modified the text and added the confidence interval of the mean to the results of the growth rates. Indeed, taking into account the changes made following the suggestions of the next comment, the GR are not significantly different. The text has been modified as follows:*

*Growth rates ranged from 2.9 to 7.6 nm $h^{-1}$ at the suburban site, with a mean value of 4.5±2.1 nm $h^{-1}$, and from 1.4 to 4.0 nm $h^{-1}$ at the urban stations with a mean value of 2.8±1.0 nm $h^{-1}$. We cannot affirm that the mean value of the suburban station is higher than that of the urban stations because the mean value of the GR at urban stations is included in the confidence interval of the GR at the suburban station. It also has to be considered that only a few days of measurements are available for this calculations. The GR calculated are consistent with those observed by Alonso-Blanco et al. (2017), ranging 1.4-10.6 nm $h^{-1}$ at CIEMAT.*

*We modified Figure 2 and also deleted the conclusions of our previous results (i.e. greater GR at the suburban station) accordingly in the abstract and conclusions.*

- Moreover, table 1 shows that the highest GR in the suburban station were measured on the 16/06 and 17/06 but from figure 1 is it evident that during these days there is not a clear growth and the GR calculation can easily be biased by the absence of a defined growing mode. The authors should comment on this and eventually revise their statements.

*REPLY: Yes, thank you for this observation. We deleted these days for the calculation of GR and recalculated the mean, percentiles and confidence intervals. The previous reply is already written having into account these changes and these days have been deleted from table 1 and Figure 2. We also revised the GR for the other days and corrected it for 14/07/2016.*

- The argumentations provided in section 3.4.2 to support the hypothesis that the airport is the source of the nocturnal UFP peak are not strong enough in my opinion. I thank the authors for preparing figure S8, this is really helpful and confirms my initial scepticism. The nocturnal UFP peak lasted for about one hour on all the 3 days, whereas the wind conditions (NE direction, strong wind speed) stay more or less constant over several hours. Moreover, Madrid airport seems to have flights during all night long so the period with high UFP concentration should last much longer. I cannot see how this UFP peaks can be linked with the airport given the information provided in this manuscript. The authors should provide more convincing argumentation in favour of this hypothesis or look for other possible causes.

REPLY: We agree with this comment and we modified our conclusions accordingly as follows:

*Although out of the major focus of this study (photochemical nucleation), other interesting events were detected taking place during night time. […]*

*Furthermore, these episodes occur outside local traffic rush hours, and are registered together with strong NE winds, which suggest that they might be transported from a stationary source and not formed locally. To better support this hypothesis, Fig. S8 shows PSM data together with wind direction and wind speed, showing that the episodes coincide with strong NE winds, .*

*[…]*

*In the discussion paper we pointed out the airport Adolfo Suárez Madrid-Barajas, located NE of the city, as a possible source of these high UFP concentrations.  However, the UFP peaks lasted for about one hour on all the three days, whereas the strong NE wind prevailed a few hours. Moreover, the airport has flights during all night, therefore a longer period with high UFP should be observed. Although other studies have linked aircraft emissions with nucleation bursts without growth (Cheung et al., 2011, Masiol et al., 2017), with this study we cannot affirm that the airport is the origin of these bursts. As mentioned before, these episodes were unexpected and were not the main focus of this study. To elucidate the origin of these UFP bursts further research will be required.*

In the abstract and conclusions, we changed this sentence*: Additionally, a few nocturnal bursts of nucleation mode particles were observed in the urban stations, which could preliminarily be related with aircraft emissions transported from the airport of Madrid.*

By This one: *Additionally, a few nocturnal bursts of nucleation mode particles were observed in the urban stations, for which further research is needed to elucidate their origin.*

I would like to stress about the importance of being careful with these kind of overstatements: this manuscript was highlighted on the Airmodus webpage (https://airmodus.com/nocturnal-sub-3-nm-particles-in-madridairmodus-newsletter-22018/) with a title saying that the airport is a source of ultrafine particles in Madrid no matter the fact that the paper was still in the review phase and no strong evidences for this causation links were provided.

*REPLY: We regret this situation but it is Airmodus who took the initiative to publish this preliminary results when they were still under discussion. We agree with you, even when the main topic of the paper is photochemical nucleation and this was a secondary issue concerning our objectives.*

In addition to these 2 major comments I also have a couple of less important considerations:

- Page 4 lines 16-20: I don't really understand what the authors mean here. They mentioned that important discrepancies were found but I don't understand how this was taken into account. The authors should rephrase this part or add a couple of sentences to explain it better.

*REPLY: We re-phrased the sentence as follows: The aerosol number size distribution was measured with an SMPS (TSI 3080) for the size range 15-660 nm and a 1 nm SMPS (TSI 3938E77) for the size range 1-30 nm.  In the overlapping range 15-30 nm, the nanoSMPS yielded slightly higher concentration values. In order to correct that, and to obtain a continuous size distribution,  the daily nanoSMPS values were corrected to adapt those of the SMPS data. We compared the resulting merged particle size distribution with CPC measurements, to check that there was a good agreement in the total particle concentration*

- Page 4 line 29-30: In my previous review I asked to provide more details about the sampling conditions, I appreciate that the authors added this sentence but I was expecting something more about the length/size of the inlets, the flow rates and the losses inside those lines. The authors should add a short description of their setup.

*Reply: We added the information of the length and size of the inlets, the flow rates and the material used to reduce losses inside those lines. The paragraph was changed from this:*

*TSI instruments at all stations were corrected for diffusion losses and multiple charge losses using the instruments' own software. The measuring conditions were the same at all sites.*

*To this:*

*TSI instruments at all stations were collocated next to windows or walls where holes were available for inlets, and equipped with individual ¼ inch, 20 cm long conductive silicone tubing inlets for PSM. SMPS and CPC also had individual 30 cm conductive silicone tubing inlets. Being the inlets individual, each instrument had its own flow rate. TSI instrument data were corrected for diffusion losses and multiple charge losses using the instruments' own software.*

Page 8 line 9: here it is said that "only 3 days of PSM data were available" however this is misleading because PSM data are available for a much longer period. What is missing is a longer overlapping between PSM and SMPS data so this sentence should be made more clear.

*REPLY: Thank you, we modified the sentence to:*

[revised manuscript text omitted]

Figure 1

[Figure]

 Figure 2

[Figure]

Figure 2

[Figure]

Figure 3

[Figure]

Figure 5a

[Figure]

Figure 5b

[Figure]

[Figure]

Figure 47

[Figure]

10   Figure 6

[Figure]

[Figure]

Figure 7

[Figure]

Figure 28

[Figure]

5    Figure 9

---

## Author Response (AR3)

**REPLIES TO REFEREE #2 COMMENTS**

**Page 1 line 37: please speak of ultrafine particles and not NPF because is not possible to know if particles are really formed over the full vertical extension of the mixed layer, as stated in chapter 3.3. The same apply to the conclusions (page 12 line 1).**

5 to the conclusions (page 12 line 1).
 We agree and the text has been modified as follows:
 This indicates that NPF occurs UFP are detected quasi-homogenously in an area spanning at least 17 km horizontally.

10 **Page 10 line 34: as in the case of the growth I would not say that that the shrinkage is faster in the suburban.** We agree and this sentence has been deleted.

**REPLIES TO EDITOR'S COMMENTS**

15

Page 2, Line 15: Define UFP here (currently it is defined in line 24 below) Page 2, Line 31: Volatile -> volatile (to be consistent) Page 3, line 35: Add "the" before "Madrid metropolitan area" Page 4, line 29: Add "to" before "study". We have made these changes as suggested.

Page 4, line 31: Please define CSIC.

We have included the definition: Consejo Superior de Investigaciones Científicas, Spanish national research council.

25

20

**Page 4, 2nd last paragraph: Here and throughout the manuscript, please harmonize the capitalization of instruments and acronyms.**

This section has been rewritten as suggested by another comment and this has been corrected.

**30 Page 4, line 38: Define CIEMAT.**

We have included the definition and English translation: *Centro de Investigaciones Energéticas, Medioambientales y Tecnológicas, Research center for energy, environment and technology.*

**Page 5, line 5: Define CPC.**

35 We have included the definition: *Condensational Particle Counter*. Note that this is now defined in a different line, since this section has been modified according to another comment.

Page 5, line 5: As addressed by the reviewers, the comparison between the different instruments is an important statement. Please quantify the agreement (e.g. with statistical values or an extra scatter plot for

the supplement). As with every experimental paper, the reader should expect more details on the set-up. Please add all needed details on the operation and set-up (incl. flows, loss estimations, calibration procedures for the each used instrumentation). These are essential experimental details. Please take care in correctly defining the instrumentation (see detailed comments below).

We have rewritten section 2.2 as follows to include details about the instrumentation and set-up:

[revised manuscript text omitted]

- 25 or walls where holes were available for inlets, and equipped with individual 1/4 inch, 20 cm long conductive silicone tubing inlets for PSM. SMPS and CPC also had individual 30 cm conductive silicone tubing inlets. Being the inlets individual, each instrument had its own flow rate. TSI instrument data were corrected for diffusion losses and multiple charge losses using the instruments' own software.
- 30 Page 5, line 9: Please define ISCIII (necessarily not known to readers outside of Spain). We have included the definition and English translation: *Instituto de Salud Carlos III, Institute of health Carlos III.*

**Page 5, line 12: Define PTR-ToF-MS. Again, a brief description of one of the key instrumentation should be given here. The reader should follow your paper without consulting extra literature.**

- We have included the definition of PTR-ToF-MS here: *proton-transfer reaction time-of-flight mass spectrometer*. However, we think it is not necessary to include a more detailed description of this instrument, since it is not one of the key instruments in this study and it is only referred to in the supplement.
- 40 Page 5, line 15: "TSI instruments" is not the correct term here (TSI is a company). Replace by a more appropriate word (e.g. "Particle sizing and counting instrumentation"). This was corrected as suggested.

**Line 17: What do you mean with "Being the inlets individual, each instrument had its own flow rate."?**

45 No common inlet+flow splitter was used for sampling. Each instrument sampled air from an individual tubing through the window. To make this clearer the following text *TSI instruments at all stations were collocated next to windows or walls where holes were available for inlets, and equipped with individual ¼ inch, 20 cm long conductive silicone tubing inlets for PSM. SMPS and CPC also had*

individual 30 cm conductive silicone tubing inlets. Being the inlets individual, each instrument had its own flow
rate. TSI instrument data were corrected for diffusion losses and multiple charge losses using the instruments' own software.

was corrected as follows:

55 Instruments for UFP measurements were collocated to sample through the window and equipped with individual  $\frac{1}{4}$  inch 20-30 cm long conductive silicone tubing inlets with a cyclone at the end to avoid particles over  $1\mu m$ . TSI

instrument data were processed and corrected for multiple charge and diffusion losses by using the TSI AIM software. PSM data were processed and corrected for losses by using Scilab code provided by Airmodus.

**Page 5, line 21: Please define "Hy-SMPS". Maybe rephrase the sentence.**

Hy-SMPS is the minituarized SMPS designed by Prof. Kang-Ho Ahn (Hanyang University, Repubic of Korea). 5 This has been clarified in the text.

**Page 5, line 26: Here and throughout the entire section 2.2, please take care in properly describing the used instrumentation. For example, it should say "The instrument was intercompared with another SMPS (TSI Inc, USA, Model 3776 with a standard XXX DMA) ...".**

Section 2.2 has been rewritten as suggested by a previous comment. This has now been corrected.

**What is a "Hy-CPC"? Is it a company or a special CPC?**

Hy-CPC is the name of the miniaturized CPC designed by the University of Hanyang. This has been clarified in the text.

15

10

**A CPC usually only counts particles. How was the range 3-1000 nm assured or did you integrate a measured size distribution? Please properly define your used instrumentation!**

This was a mistake. As you stated, CPC counts particles, in this case particles larger than 3 nm. We have corrected the text as follows: 20

Number concentration of particles larger than 3 nm was measured with a miniaturized butanol-based CPC (Hy-CPC, designed by the University of Hanyang).

Page 6, line 3: To be consistent, you could consider using the recently defined acronym for growth rate (GR). We have changed it according to your suggestion. 25

**Eq. 3: Please define alpha.**

We have included the definition:  $\alpha$  is the sticking coefficient, here assumed to be equal to 1, as in most studies.

**Eq. 5: Please properly define the difference between Dp and Dp'.**

There is no formal difference between Dp and Dp' here. We used this notation to differentiate between the full size 30 range considered for the calculation, and the additions of the summation, which correspond to each size bin measured.

**Page 6, line 32: Suggest to remove the "s" after balloons.**

We have removed the "s" as suggested. 35

> Page 7, line 12: Add "number" before "particle" (since you show a particle number size distribution) We have corrected this as suggested.

Page7, line 19: I would suggest to rephrase the beginning of this sentence to: "Since all stations are differently 40 influenced by traffic ..." Please specify what you mean with "Otherwise, we would observe significant differences at our stations." It is not really clear what you mean here.

To clarify this, the following text:

Since all stations are differently influenced by traffic, we can affirm that these episodes are regional events and not 45 representative of primary emissions. Otherwise, we would observe significant differences at our stations. Was corrected as follows:

If the episodes were caused by primary emissions, then we would observe different size distributions at all stations, because each one of them is differently influenced by traffic. The urban station is largely influenced by traffic emissions, whereas the suburban station is much less affected by these emissions. Since we observe the same size distribution at both stations, then we can say that traffic emissions are not the origin of the observed distribution.

50

**Figure 1: Please add proper and correct y-axis labels to your figures (incl. the colorbar) and remove your internal acronyms (I assume you show particle diameter).**

We have modified the axis and colorbar labels accordingly to include variables and units and deleted the stations acronyms. The figure caption now contains the information of the stations corresponding to each plot. 55

**Figure 8: Why is there only one 25/75th percentile value shown for the urban / Dp>40nm values?**

Both 25 and 75th percentiles are shown (limits of the box). What is missing is the whisker below the 25th percentile, but there is one outlier, meaning that there is only one value below the 25th percentile. This is due to the limited number of points used in this group.

5 Conclusions: The acronyms 'CS' and 'CoagS' are defined again but then not used. I suggest removing them. We agree and they have removed.

Figure S2: Why do you have doubly charged particles also to the left of the main peak at 50nm?

10 Figure R1: Schematic of the experimental set-up for the intercomparison between TSI-SMPS (Standard DMA with 3776 CPC) and Hy-SMPS.

PARTICLE DIAMETER(nm)

25

Figure R2: Normalized number distribution of extracted particles by the DMA.

15 After neutralization (first step of the set-up shown in Fig. R1), the singly charged and doubly charged particles in Fig. R2 reach a new Boltzmann energy level, i.e., the singly charged particles have 0 charge and +1, -1 charges, +2, -2 charges and so on. The doubly charged particles will also have 0 charge and +1, -1 charges, +2, -2 charges and so on.

The second DMA (in this case SMPS) scans the particle size distribution, i.e., single charged particles (central peak in Fig. S2), 2e charged particles, previously singly charged (left peak in Fig. S2) and e charged particles, previously

20 in Fig. S2), 2e charged particles, previously singly charged (left peak in Fig. S2) and e charged particle doubly charged (right peak in Fig. S2).

In general, I think you should revise once more if not certain supplementary material should be moved to the main manuscript (e.g. S1). There are a many references to supplementary material inside the text which made me switch back and forth between main manuscript and supplement. It reads not very well if a new section start with referencing to the supplement (see e.g. Sect 3.1).

Following your suggestion, we have now moved S1 and S5 from the supplement to the main manuscript to make it more readable.

**Vertical and horizontal distribution of regional new particle**

**formation events in Madrid**

Cristina Carnerero1,2, Noemí Pérez1, Cristina Reche1, Marina Ealo1, Gloria Titos1, Hong-Ku Lee3, Hee-Ram Eun3, Yong-Hee Park3, Lubna Dada4, Pauli Paasonen4, Veli-Matti Kerminen4, Enrique Mantilla5, Miguel Escudero6, Francisco J. Gómez-Moreno7, Elisabeth Alonso-Blanco7, Esther Coz7, Alfonso Saiz-Lopez8, Brice Temime-Roussel9, Nicolas Marchand9, David C. S. Beddows10, Roy M. Harrison10,11, Tuukka Petäjä4, Markku Kulmala4, Kang-Ho Ahn3, Andrés Alastuey1, Xavier Querol1

[revised manuscript text omitted]

---

## Author Response (AR4)

**REPLIES TO EDITOR'S COMMENTS**

**In general, the comments were sufficiently addressed. However, I still see a major issue in the proper usage of English language (e.g. sentences like "Being the inlets individual, each instrument had its own flow rate." do not make any sense). I therefore strongly urge you or one of your co-authors (there seem to be some native speakers) to do one more proper editorial read before it can be finally accepted. Especially revise the grammar and wording of Sect. 2.2. but please don't neglect the rest of the manuscript.**

A thorough proofreading has been done to all the manuscript. Language corrections were made in all sections, especially in Sect. 2.2. We believe that the quality and readability of the text has improved significantly.

**Concerning your whisker plots (Fig 4 and 10), please also add the number of data points for each case.**

The number of data points have been added to the figures and this has been indicated in the corresponding figure captions.

[revised manuscript text omitted]

Figure 1

[Figure]

Figure 2

[Figure]

Figure 3

[Figure]

Figure 4

[Figure]

Figure 5

[Figure]

Figure 6

[Figure]

Figure 7

[Figure]

Figure 8

[Figure]

5    Figure 9

[Figure]

Figure 10

[Figure]

5    Figure 11

---

## Author Response (AR5)

**Replies to editor's comments**

1. **Thank you for your revised manuscript. The English language has improved. However, in your revised figures 4 and 10, you now show that you have only a limited amount of data on which your conclusions are based on. A box-and-whisker plot with only three data points is highly questionable. Hence, I would like that you (together with your co-authors) revise your statistical analysis and figures once more.**

Thank you for your review. We agree that the amount of data is limited, especially regarding the shrinkage rates. Nevertheless, we believe that the main results of the manuscript are those regarding the vertical distribution of UFP. Both the GR analysis, and especially the SR analysis, are complementary results in the manuscript. We believe that we should delete the boxplot figures (figures 4 and 10) as you pointed out, but we think that the short discussions that accompanied the figures is still valid.

Regarding GR, it was already stated in the text that the differences observed between urban and suburban stations are not significant taking into account the uncertainties, which are also indicated in the manuscript. We have kept the observations in Table 1, and we have emphasized in the fact that there is only a limited number of observations and that further research is needed to confirm our observations. The text has been modified as follows:

For the observed daily regional NPF events, GR for the nucleation mode, $J_1$, CS, and CoagS have been determined using PSM and SMPS aerosol size distribution measurements. Here, the GR is calculated from the time of detection of the smallest mode until either the particle reaches 25 nm or it stops growing before reaching that size. We considered only the events that are observed simultaneously at the suburban station and at least at one urban station (highlighted in Table 1).  GR regarding the vertical measurements are discussed in the following section due to differing sampling periods.

The calculated GR for the surface stations are shown in Table 1. GR ranged from 2.9 to 7.6 nm h$^{-1}$ at the suburban site, with a mean value of 4.5±2.1 nm h$^{-1}$, and from 1.4 to 4.0 nm h$^{-1}$ at the urban stations, with a mean value of 2.8±1.0 nm h$^{-1}$. We cannot affirm that the mean value of the suburban station is higher than that of the urban stations because the mean value of the GR at urban stations is included in the confidence interval of the GR at the suburban station. It also has to be considered that only  seven days of measurements are available for these calculations, hence further studies would be needed to confirm the observed differences between urban and suburban stations. Nevertheless, the GR calculated are consistent with those observed by Alonso-Blanco et al. (2017), ranging 1.4-10.6 nm h$^{-1}$ at CIEMAT.

Regarding SR, we have removed the boxplot but we considered that it would be interesting to maintain the short description of the observed episodes. We have emphasized in the fact that we are describing simultaneous episodes in different stations, hence the limited amount of data. We have also given emphasis to the need of further research in order to confirm the results. The corresponding text has been modified as follows:

The start of the shrinking phase coincides with a marked increase in wind speed (Fig. S5); therefore, it is associated with dilution, which favors the evaporation of semi-volatile vapors, resulting in a decline in particle diameter and concentrations, as observed in most cases.  The calculated shrinkage rates are shown in Table S1.  SR for particles with a starting diameter below 40 nm range from -1.1 to -8.0 nm h$^{-1}$. For particles in the Aitken mode above 40 nm, the values fall between -4.9 and -20.5 nm h$^{-1}$. The results  seem to indicate that particles shrink faster the larger the starting diameter is. Since shrinkage is observed simultaneously at urban and suburban stations,  shrinkage  seems to be a regional phenomenon in the Madrid area, as already suggested by Alonso-Blanco et al. (2017).  However, we could only identify a limited number of simultaneous shrinking episodes, and further research would be needed to confirm these results.

**2. Please add error bars to Fig. 5.**

We have added 95% confidence intervals (CI) to the data in Figure 5 (now Fig. 4), which now provide an estimation of the uncertainties taking into account the number of data points considered (see figure A1).

$CI = \bar{x} \pm 1.96\, SE$

where $\bar{x}$ is the mean value of $x$, and SE is the standard error of the mean:

$$SE = \frac{\sigma}{\sqrt{N}}$$

where $\sigma$ is the standard deviation, and N the number of data points considered.

We think that the conclusions that we had previously stated concerning this figure are still valid taking into account the uncertainties and the limited number of observations.

[revised manuscript text omitted]

Figure 1

[Figure]

Figure 2

[Figure]

5    Figure 3

[Figure]

Figure 4

[Figure]

Figure 5

[Figure]

Figure 6

[Figure]

Figure 7

[Figure]

5    Figure 8

[Figure]

Figure 9